

# Discontinuity of the concentration and composition of dissolved organic matter at the peat-pool interface in a boreal peatland

Antonin Prijac [1,2,3], Laure Gandois [4], Laurent Jeanneau[6], Pierre Taillardat[1,7], Michelle Garneau[1,2,5]

[1] Centre de Recherche sur la Dynamique du Système Terre (GÉOTOP), Université du Québec à Montréal, Canada
[2] Groupe de Recherche Inter-universitaire en Limnologie (GRIL), Université du Québec à Montréal, Canada
[3] Institut des Sciences de l'Environnement (ISE), Université du Québec à Montréal, Canada
[4] Laboratoire d'Écologie Fonctionnelle et Environnement, UMR 5245, CNRS-UPS-INPT, Toulouse, France
[5] Département de Géographie, Université du Québec à Montréal, Canada
[6] Laboratoire Géosciences Rennes, UMR 6118, CNRS-Université de Rennes, France
[7] Integrated Tropical Peatland Research Program (INTPREP), National University of Singapore, Singapore

Correspondence to: Antonin Prijac (prijac.antonin@courrier.uqam.ca)

**Abstract**

Pools are common features of peatlands and can represent from 5 to 50% of the peatland's surface area. They play an important role in the peatland carbon cycle by emitting carbon from their surfaces to the atmosphere. However, the origin of this carbon is not well known. A hypothesis is that carbon emitted from pools is the product of mineralised peat-derived dissolved organic matter (DOM). To test this hypothesis, this study examined the origins, compositions, and degradability of DOM in peat porewaters and peat pools within an ombrotrophic boreal peatland in northeastern Québec (Canada) for two years during the growing season. The temporal evolutions of dissolved organic carbon (DOC) concentrations, the optical properties, molecular compositions (TMH-GC-MS), stable isotopic signatures ($\delta^{13}$C-DOC), and degradability of DOM were determined. This study demonstrates that DOM is a complex and highly dynamic component of peatland ecosystems. If the molecular analyses reveal that DOM in porewaters and pools share a common vegetation origin, the compositions of the DOM in the two environments are markedly different. Peat porewater DOM is more aromatic, with a higher molecular weight DOC:DON ratio. The temporal dynamics of DOC concentrations and DOM compositions also differ. In peat porewaters, the DOC concentrations followed a strong seasonal increase from 9 mg L$^{-1}$, reaching a plateau above 20 mg L$^{-1}$ in summer and autumn. This is best explained by seasonal vegetation productivity, which is greater than DOM degradation through microbial activity. In pools, DOC concentrations also increased but remained two times lower than in the peat porewaters at the end of the growing season with an average concentration of 10 mg L$^{-1}$. Those differences might be explained by a combination of physical, chemical, and biological factors. The limited hydraulic conductivity in deeper peat increased the DOM residence time in peat. This might favour both DOM microbial transformation within the peat and the interaction of DOM aromatic compounds with the peat matrix, explaining part of the shift of DOM compositions between peat porewaters and pools. The DOM might be even further transformed at the interface between peat and pools with the production of low molecular weight compounds. This study could not highlight any photolability of DOM and only limited microbial degradability. We estimate that the carbon emissions related to DOM transformation in peatland pools could represent from 4.2 to 8.7% of the long-term apparent rate of carbon accumulation.





## 1. Introduction

Northern peatlands constitute one of the most important terrestrial reservoirs of organic carbon (C) containing about $530 \pm 160$ Pg C while only covering less than 3% of the global terrestrial land surface (Yu, 2012; Loisel et al., 2017; Xu et al., 2018). The net ecosystem carbon accumulation rates of peatlands are typically greater than the losses to the atmosphere through peat degradation and lateral transfer (Billett et al., 2006, 2012; Blodau et al., 2007; Tunaley et al., 2017). Peatlands are characterised by a mosaic of surface microforms, such as hummocks, laws, hollows, and pools (Harris et al., 2020). Considering peatlands as a patchwork of microforms rather than a homogeneous ecosystem is critical to accurately quantify their carbon balance and the role they play in the modern global carbon cycle. Carbon dynamics between microforms are closely related to the vegetation type and water table depth which influence the carbon dioxide ($CO_2$) and methane ($CH_4$) exchange with the atmosphere (Nungesser, 2003; Chaudhary et al., 2018). Among the different microforms, pools can constitute an important proportion of peatland surface areas, ranging from 5 to 50% (White, 2011; Pelletier et al., 2014, 2015) and represent a net carbon source to the atmosphere (Pelletier et al., 2014). While most studies of peatland carbon dynamics have focussed on terrestrial microforms (Nungesser, 2003; Pelletier et al., 2011; Shi et al., 2015; Chaudhary et al., 2018; Graham et al., 2020), the composition and processes of organic carbon in pools remain poorly documented.

The composition of dissolved organic matter (DOM) has been documented in peatland porewaters. A complex mixture of compounds with a diversity of compositions, functional groups, and ages seem to co-exist (Tipping et al., 2010; Kaplan and Cory, 2016; Raymond and Spencer, 2015; Tiwari et al., 2018; Dean et al., 2019; Tfaily et al., 2018). The production of DOM in peat porewaters is controlled by vegetation productivity, peat temperature (Rydin et al., 2013; Kane et al., 2014), and microbial activity (Worrall et al., 2008).

It has also been shown that pools can represent active compartments of peatland ecosystems for DOM (Laurion and Mladenov, 2013; Deshpande et al., 2016; Payandi-Rolland et al., 2020; Folhas et al., 2020; Laurion et al., 2021) – a topic that has been less well studied. The DOM of pools may derive from surrounding terrestrial peat (i.e., allochthonous) or be the result of their internal primary production (i.e., autochthonous). In both peat porewaters and pools, DOM is affected by biodegradation processes and by photodegradation in pools (Lapierre and del Giorgio, 2014; Vonk et al., 2015). Changes in DOC concentrations and DOM composition are commonly observed and associated with a wide range of degradation rates (Frey et al., 2016; Payandi-Rolland et al., 2020; Moody and Worrall, 2021). The composition and reactivity of DOM transferred from the terrestrial to aquatic compartments of peatlands highly depend on the hydrology and hydroclimatic context, and biological and chemical processes occurring during their transfer (Jaffé et al., 2012; Kaplan and Cory, 2016). The DOM transfers between peatlands and aquatic ecosystems are well documented for streams (Elder et al., 2000; Billett et al., 2006, 2012; Austnes et al., 2010; Knorr, 2013; Frey et al., 2016; Buzek et al., 2019; Dean et al., 2019; Rosset et al., 2019) but more rarely for pools (Banaś, 2013; Arsenault et al., 2019; Payandi-Rolland et al., 2020).

Differences in DOM composition and concentrations between peat porewaters and pools have been observed, but the processes involved remain unclear (Schindler et al., 1997; Payandi-Rolland et al., 2020). Studies conducted in temperate peatlands have highlighted that the morphology (e.g., size, shape, depth, slope of banks) and surrounding vegetation influence the carbon contents and hydrochemistry in the pools (Banaś, 2013; Arsenault et al., 2018, 2019).



Others have explained the changes in DOM composition as the result of photodegradation and biodegradation in pools (Laurion and Mladenov, 2013; Arsenault et al., 2019; Laurion et al., 2021). Studies investigating the changes in DOM composition in peatland porewaters and pools have mostly been focused on temperate (Banaś, 2013; Arsenault et al., 2019), subarctic, and Arctic regions (Laurion and Mladenov, 2013; Deshpande et al., 2016; Payandi-Rolland et al., 2020; Laurion et al., 2021; Moody and Worrall, 2021), but there is no insight about changes in DOM compositions in boreal peatlands.

Because DOM may derive from many different sources and be subjected to various processes of transformation and degradation, apprehending the complexity of the origins, compositions, and properties of the molecules that comprise the DOM is challenging. The use of complementary analytical methods is a good approach to characterise DOM and attempt to understand its origins and composition (Folhas et al., 2020; Tfaily et al., 2013). The DOC:DON elementary ratio is used to estimate the microbial processing of DOM (McKnight et al., 1994; Autio et al., 2016). Absorbance and fluorescence are recognised tools to estimate the average DOM molecular weight and aromaticity (Haan and Boer, 1987; Weishaar et al., 2003; Helms et al., 2008), discriminate the origins of DOM between microbial and plant sources (McKnight et al., 2001; Cory et al., 2010), and highlight microbial degradation (Parlanti, 2000; Wilson and Xenopoulos, 2009). The stable carbon isotopic signature of DOM can be used to discriminate between plant-derived and processed DOM (Elder et al., 2000; Billett et al., 2012; Buzek et al., 2019). Finally, the analysis of target molecules using TMH-GC-MS allows the definition of indicators of DOM sources and processing (Jeanneau et al., 2014, 2015; Kaal et al., 2017, 2020).

The aim of this study is to investigate the role that pools play in the production, transfer, and transformation of organic carbon within peatland ecosystems. We identified three possible scenarios. First, pools are mineralisation hotspots that can decompose the laterally exported fresh organic matter from adjacent peat porewaters. Second, pools represent passive pipes that only collect the remaining refractory DOM laterally exported from peat porewaters. Third, pools represent a sub-ecosystem within the peatland where both primary productivity and heterotrophic respiration exist. To clarify which scenario is the most accurate, we studied the spatiotemporal variability of DOM over two growing seasons. The objectives of this study were to: i) identify the differences in the origin, quantity, composition, and degradability of DOM between peat porewaters and pools; ii) understand which factors induce those differences; and iii) discuss the contribution of DOM in pools to the peatland carbon budget.

## 2. Study site

The study site is located in northeastern Quebec, Canada, within the Romaine River watershed (14,500 km$^2$), adjacent to the Labrador border. It is located in the eastern spruce-moss bioclimatic domain of the closed boreal forest (Payette, 2001) at the limit of the coastal plain and the Highlands of the Laurentian Plateau of the Precambrian Shield (Dubois, 1980). The Bouleau peatland (unofficial name; 50°31'N, 63°12'W; alt: 108 ± 5 m) is an ombrotrophic, slightly dome-shaped bog with a total surface area of 1.5 km$^2$. Peat accumulation was initiated at ca. 9260 cal BP and the maximum peat depth reaches 440 cm (Primeau and Garneau, 2021). The mean annual temperature is 1.5°C and the mean annual precipitation is 1011 mm, of which 590 mm falls as snow. The average monthly positive temperatures occur from



May to October with 1915 growing degree days above 0°C (Havre-Saint-Pierre meteorological station, mean 1990-2019; Environment of Canada, 2019).

The surface microforms of the Bouleau peatland show a clear patterned surface of alternating dry hummocks, lawns, hollows, and pools. The surface vegetation varies according to the microtopography with *Sphagnum fuscum*, *S. capillifolium*, and *Cladonia rangiferina* on hummocks, *S. magellanicum*, *S. rubellum*, *S. cuspidatum*, and *Trichophorum cespitosum* on lawns, and *S. majus* and *S. pulchrum* on hollows (Primeau and Garneau, 2021). Pools cover 9% of the peatland surface area and are characterised by their elliptical morphology, steep banks, and slightly

concave bottoms. Because of the steep banks, no aquatic vegetation is observed along the edges of the pools.



**Figure 1.** Aerial view of the Bouleau peatland with the location of the sampling sites (green dots = wells for peat porewater; yellow triangles = pools; blue diamond = stream outlet).



## 3. Material and methods

### 3.1. Water sampling

Sampling was performed five times during the 2018 growing season (June, July, August, September, and October) and four times in 2019 (June, August, September, and October). In 2018, six pools were sampled (M1 to M6, Fig. 1) and in 2019, five additional pools were included (M11 to M15, Fig. 1). The pool sizes vary from 30 to 2065 m$^2$ with a mean depth varying between 70 and 120 cm. The pool water was sampled from the banks at the surface of the water column.

The peat porewater was sampled from six wells (P1 to P6, Fig. 1) located along a topographic transect from the dome to the southern edge of the peatland. Two-metre-long PVC wells were perforated and covered with a nylon sock to avoid infilling by peat. They were inserted in peat to collect water in the first two metres of the peat column.

The physicochemical parameters (temperature, pH, specific conductivity, and dissolved oxygen saturation) of peat porewater and pool water were measured using a multi-parameter portable meter (Multiline Multi 3620 IDS, WTW, Germany) at each sampling site and calibrated before each field visit. All water samples were collected in clean polypropylene (PP) bottles (acid rinsed and pre-combusted for 4h at 450°C) and filtered on pre-combusted (4h at 450°C) GF/F filters (Whatman).

### 3.2. Water level and temperature monitoring

#### 3.2.1. Field instrumentation

The six wells were equipped with a water level data logger U20-001-04 in 2018 and replaced with a U20l-04 in 2019 (HOBO, Onset, USA) for continuous measurements of the water table depth (WTD) and temperature from June 2018 to October 2020. Water temperature was recorded hourly in pools M11 to M15 using HOBO TMC50 probes coupled with a HOBO U12-008 data logger (Onset, USA) from June 2019 to August 2020. A water level data logger was installed in pool M11 (Fig. 1) and water level variations were measured from May 20$^{th}$ to August 28$^{th}$ 2020. Height variations (in cm) between the peatland surface at wells P5 and P6 and adjacent pool M11 were measured using a Zip Level Pro-2000 (Technidea, USA). Those measurements allowed the water levels in the pool to be compared with those in the two wells (Fig. SI.2). An EXO2 multiparameter probe (YSI, USA) was installed at the outlet of the peatland stream to record, among others, water temperature hourly, from June 2018 to August 2020.

#### 3.2.2. Definition of the seasons

Samples from the two studied years were pooled according to seasons. In this study, seasons were defined based on air and water temperatures measured at the site (Fig. SI.3). Spring was defined from the end of the seasonal thaw that occurred in May to the end of June. In spring, the mean air temperature was 12.7 ± 2.1°C while the mean water temperatures from peat porewaters and pools were 12.3 ± 5.2°C and 14.8 ± 1.7°C, respectively. During spring, ice lenses were still observed within some peat layers and along the pool edges with less sun exposure. Summer included the months of July and August when air and water temperatures were at their warmest (respectively 17.2 ± 1.5°C in July and 16.4 ± 1.4°C in August for air temperatures; 16.7 ± 3.8°C and 16.30±1.6°C in peat porewaters and 21.1 ±





1.8°C and 23.4 ± 3.0°C in pools). Finally, the autumn season corresponded to the months of September and October
when air and water temperature decreased to zero. September was characterised by a moderate decrease of water
temperatures to 15.7 ± 1.0°C in peat porewaters and 17.4 ± 1.8°C in pools, but a drastic decrease in mean air
temperatures with an average of 9.5 ± 2.5°C. October was the coldest sampling period for both air temperature (4.3 ±
2.1°C on average) and water temperatures in peat porewaters (8.25 ± 2.5°C) and in pools (8.10 ± 1.8°C).

**3.3.    Quantitative analyses**

The filtered water samples were prepared for DOC and total nitrogen (TN) analyses by acidification to pH 2 with 1M
HCl and stored in 40mL glass vials. The DOC and TN concentrations were analysed using the catalytic oxidation
method followed by the non-dispersive infrared (NDIR) detection of the $CO_2$ produced (TOC analyser TOC-L,
Shimadzu, Japan) with limits of quantification of 0.1 mg C $L^{-1}$ and 0.2 mg N $L^{-1}$.

The samples were prepared for cation and anion analyses and stored in high-density polyethylene (HDPE) vials
without acidification. These ions (chloride, ammonium, nitrites, and nitrates) were analysed by high performance
liquid chromatography (HPLC) coupled with a Dionex ICS-5000+ analyser for anions (Thermo Fisher Scientific) and
a Dionex DX-120 analyser for cations (Thermo Fisher Scientific).

The reference materials included ION-915 and ION 96.4 (Environment and Climate Change Canada, Canada). The

analyses were performed at EcoLab (UMR 5245 CNRS – UT3 – INPT, France).

Dissolved organic nitrogen (DON) corresponds to the difference between the concentration of TN and the sum of
concentration of inorganic nitrogen (ammonium, nitrites, and nitrates).

**3.4.    Qualitative analyses**

**3.4.1.    Stable isotopic analyses**

Analyses of $\delta^{13}$C-DOC were realised at the Jan Veizer stable isotope laboratory (University of Ottawa, Canada)
following the method developed by Lalonde et al. (2014). The samples were acidified to pH 2 with 1M HCl and stored
in 40mL quality certified ultra-clean borosilicate glass vials. The first step involved the catalytic oxidation of DOC
followed by a solid-state non-dispersive infrared (SS-NDIR) detection of the $CO_2$ produced (OI Aurora 1030C, Xylem
Analytics, USA). The produced $CO_2$ was passed through a chemical trap and a Nafion trap prior to $^{13}$C isotopic

analyses using isotope-ratio mass spectroscopy (IRMS, Thermo Finnigan DeltaPlus XP, Thermo Electron
Corporation, USA). The results were standardised with organic standards (KHP and sucrose) and the $^{13}$C/$^{12}$C ratios
were expressed as per mil deviations from the international standard VPDB.

**3.4.2.    Optical and fluorescence analyses**

The samples for UV-visible spectroscopy analyses were stored at 4°C in glass vials following filtration on GF/F filters.

Absorbance was measured from 180 to 900 nm with a 5 nm resolution. In 2018, the absorbance analyses were
performed on Ultrospec 3100 (Biochrom, United Kingdom) over a wavelength range from 190 to 900 nm at 2 nm
intervals while in 2019 they were performed on Duetta (Horiba, Japan). All analyses were realised at the Groupe de
recherche interuniversitaire en limnologie (GRIL, UQAM, Canada). For comparison, ten samples from the 2019



campaign were randomly selected and analysed on both equipments, Duetta and Ultrospec 3100. A pairwise t-test revealed slight but significant differences between absorbance at 254 nm from the two series (t = -3.9013, df = 9, p-value = 0.0036).

Absorbance indices were calculated to provide information about DOM composition. Those indices were $SUVA_{254}$ (L $mg^{-1}$ $m^{-1}$) which is a proxy of the of DOM's aromatic content Weishaar et al., 2003), E2:E3 ratio, and spectral slope ratio ($S_R$) which are proxies of the average DOM molecular weight (Haan and Boer, 1987; Helms et al., 2008).

In 2019, spectrofluorometric analyses were also conducted on Duetta (Horiba, Japan) at the GRIL laboratory. Samples were excited at a range from 230 to 450 nm (at 2 nm resolution) and fluorescence was measured at a range from 240 to 600 nm (at a 5 nm resolution). Prior to the analyses, the samples were diluted when necessary to maintain an absorbance intensity at 254 nm below 0.6. A blank sample with MilliQ water (Merck-Millipore, Germany) was measured prior to the sample analyses. The sample spectra were obtained by subtracting the blank spectra to eliminate the Raman scatter peak. The operation was conducted automatically by the analytical equipment.

Two indices were calculated to provide qualitative information on the fluorescent fraction of the DOM: 1) the fluorescence index (FI), lower values (FI ≈ 1.4) of which indicate a plant origin while higher values (FI ≈ 1.9) indicate a microbial origin of DOM (Cory et al., 2010; McKnight et al., 2001); 2) the β:α index, which is known as a proxy of biological activity, and an increase in the ratio of which corresponds to an increasing proportion of recently produced DOM derived from microbial activity (Parlanti, 2000; Wilson and Xenopoulos, 2009).

### 3.4.3. THM-GC-MS analyses

Thermally assisted hydrolysis methylation-gas chromatography-mass spectrometry (THM-GC-MS) was performed on 37 samples from peat porewaters (n = 18) and pools (n = 19). Those samples were selected to include summer and autumn 2018 and spring, summer, and autumn 2019. The THM-GC-MS analyses were conducted on freeze-dried samples from 100mL of water previously filtered on GF/F filters (Whatman) and followed the procedure described by Jeanneau et al. (2015). One mg of sample was introduced into an 80 µL stainless steel reactor with an excess of tetramethylammonium hydroxide (6 mg). The THM reaction was performed at 400°C using a vertical microfurnace pyroliser PZ-2020D (Frontier Laboratories, Japan). The reaction products were injected into a gas chromatograph GC-2010 (Shimadzu, Japan) equipped with a SLB 5MS capillary column in split mode (60 m × 0.25 mm ID, 0.25 µm film thickness). The compounds were detected with a mass spectrometer QP2010+ (Shimadzu, Japan) operating in full scan mode. Analyses were realised at the Geosciences Rennes laboratory (UMR 6118 – Univ. Rennes – CNRS, France).

For each chromatogram, the compounds were identified based on known *m/z* ratios (Table SI.1) through comparison with the NIST library. The area of each compound was integrated for each *m/z* and corrected by a mass spectra factor (MSF). The MSF corresponds to the reciprocal of the integrated fragment proportion and the entire related fragmentogram in the NIST library. The relative proportion of each compound was calculated by dividing the compound area (for all cumulated peaks) with the sum of total integrated compound areas and expressed as a percentage.



All compounds were classified into five groups and their relative proportions were calculated: %CAR of carbohydrates
compounds (derived from both plant and microbial metabolism), %LMW_FA for low molecular weight fatty acids
(derived from microbial metabolism), %SOA for small organic acids, %HMW_FA for high molecular weight fatty
acids, and %PHENOLS for phenol markers (derived from plant metabolism). The indices were calculated for each
sample, derived from molecular analyses, and presented as in Jeanneau et al. (2015). The C/V ratio corresponds to the
sum of coumaric and ferulic acids divided by the sum of vanillic acid, vanillaldehyde, and acetovanillone. The
deoxyC6:C5 ratio is a mixing model based on the proportion of deoxyC6 carbohydrates (derived mainly from
microorganisms) and the proportion of C5 carbohydrates (derived mainly from plants). Values close to 0.5 suggested
a dominant contribution of plant-derived DOM while values close to 2 corresponded to the contribution of microbial-
derived DOM (Rumpel and Dignac, 2006). The last index corresponds to the proportion of plant-derived markers,
$f$VEG, which is the difference between the total markers and the microbial-derived markers, $f$MIC. The $f$MIC
corresponds to the proportion of microbial carbohydrates multiplied by the total proportion of carbohydrates, summed
by the proportion of microbial fatty acids, and multiplied by the total proportion of fatty acids. The MIC:VEG index
corresponds to the ratio of microbial-derived markers divided by the proportion of plant-derived markers.

### 3.5. Incubation of dissolved organic matter

#### 3.5.1. Experimental design

Dissolved organic matter from peat porewaters and pools was incubated during three sampling periods in 2019, from
June 7[th] to 13[th], July 31[st] to August 7[th], and September 4[th] to 10[th]. An incubation time of six days had to be adjusted to
seven days during the last campaign due to logistical constraints. Pool M11 was used to monitor the water level using
a barometric pressure sensor and was also sampled for incubations. The peat porewater samples consisted of a mix of
equal water volumes between five different wells to avoid variability in the well water composition.

The incubation experiments were performed on 100 ml of water filtered on GF/F filters (F) and in unfiltered (UF)
conditions. Amber glasses were used to test biodegradation (BIO) only and transparent vials were used for bio and
photodegradation (BIO+PHOTO). Each condition was incubated in triplicates. Considering the absence of
standardised incubation media between porewaters and pools (Vonk et al., 2015), biodegradation could be dependent
on the abundance of microorganisms in the samples from each environment and was expected to be greater in peat
porewater samples.

For *in situ* incubations (IS), the peat porewater samples were placed 1-2 cm below the water surface at the outlet of
the peatland (Fig. 1), where water temperature was recorded hourly with the EXO2 probe. The pool samples were
placed 1-2 cm below the water surface of pool M11 (Fig. 1). For controlled conditions (CC), the vials were placed in
a dark room in a laboratory space at Havre-Saint-Pierre where temperature was maintained between 18 and 20°C and
controlled twice each day. Both *in situ* and controlled conditions started the same day.

#### 3.5.2. Post-incubation analysis

In the end, samples incubated under UF conditions (n = 18) were filtered on a GF/F filter to analyse only the dissolved
fraction. All samples (n = 36) were prepared for DOC/TN, inorganic N, and absorbance analyses, before and after the


incubation experiments. The apparent removal rate of dissolved organic carbon (RDOC), expressed in mg day$^{-1}$,

corresponds to the amount of DOC removed during incubation, reported per day, and calculated following Equation (1).

Eq. (1)  $RDOC\ (mg\ day^{-1}) = \left([DOC]_{pre-incubation} - [DOC]_{post-incubation}\right)/incubation\ time$

The degradation rates correspond to the proportion of DOC lost per day of incubation and are expressed in %C day$^{-1}$ according to Equation (2).

Eq. (2)  $Degradation\ rate\ (\%\ C.day^{-1}) = \left.\dfrac{\dfrac{\left([DOC]_{pre-incubation} - [DOC]_{post-incubation}\right)}{[DOC]_{pre-incubation}} \times 100}{}\right/ incubation\ time$

[DOC]$_{pre-incubation}$ (mg L$^{-1}$): DOC concentration at the beginning of incubation

[DOC]$_{post-incubation}$ (mg L$^{-1}$): DOC concentration at the end of incubation

Changes in the DOC:DON ratio and absorbance indices were determined in proportion to the initial values per day for the variable $i$ following Equation (3).

Eq. (3)  $\Delta i\ (day^{-1}) = \dfrac{i_{post-incubation} - i_{pre-incubation}}{i_{pre-incubation}}/incubation\ time$

$\Delta i$: change of the variable $i$ during the incubation

$i_{pre-incubation}$: initial value of the variable $i$ at the beginning of incubation

$i_{post-incubation}$: value of the variable $i$ at the end of incubation

### 3.6.    Statistical analyses

All statistical tests were performed on R (CRAN-Project) through the RStudio interface (RStudio inc., USA) and all figures were realised with the package ggplot2 (Wickham, 2016).

Comparisons of variance tests were performed and in the next sections, the mention of significant differences refers to statistical tests using the following method. First, normal distribution was tested using the Shapiro and Wilk test, and normal distribution was considered true when the p-value was >0.05. If the distribution was not normal, a Kruskal

and Wallis test was performed to compare the averages and significant differences were considered true when the p-value was <0.05. Dunn tests were performed as *post-hoc* pairwise comparison tests to determine which group was significantly different (when the p-value <0.05). Second, the homogeneity of variance was tested using the Levene test and was considered true when the p-value was >0.05. If the homogeneity of variance was not true, Welsh ANOVA was performed, and significant differences were admitted when the p-value was <0.05. Estimated marginal means

tests were performed as *post-hoc* tests to determine significantly different groups (p-value <0.05). In cases where the normal distribution and homogeneity of variances were true, an ANOVA was performed, and significant differences were true when the p-value was <0.05. When there were significant differences, the Tukey tests were performed as *post-hoc* tests to determine which groups were significantly different (when the p-value <0.05). The results of the statistical tests are summarised in Table SI.2.

Principal component analyses (PCA) were used to explore relationships between DOM qualitative variables in peat porewater and pools. The selected variables were quantitative variables as DOC concentrations and qualitative variables as the DOC:DON ratio, optical indices (SUVA$_{254}$, E2:E3 ratio, and S$_R$), and molecular indices (deoxyC6:C5, $f$VEG, $f$MIC, MIC:VEG ratio, C/V ratio, and Ac:Al(V) ratio), as well as molecular compound proportions (%SOA,



%CAR, and %CAR_MIC, %LMW_FA, %HMW_FA, and %PHENOLS). Environmental and seasonal variables were
used as supplementary qualitative variables. Prior to PCA, a correlation matrix was performed to identify strong
correlations between the variables (Fig. SI.1). One of the correlated variables was excluded from PCA when the
correlation was >0.90 or <-0.90, with p-values <0.05. Therefore, the DOC:DON ratio (DOC:DON ~ DOC, cor = 0.90,
p-value <0.0001), %CAR_MIC (%CAR-MIC ~ deoxyC6:C5 ratio, cor = 0.99, p-value <0.0001), and $f$MIC ($f$MIC ~
MIC:VEG ratio, cor = 0.98, p-value <0.0001) were excluded from the PCA dataset. The PCA was performed with the
package FactoMineR (Lê et al., 2008).

## 4.  Results

### 4.1.      Hydrodynamics and physicochemical characteristics

Pool and peat water levels followed the same seasonal trend, although the water level in the pools was always lower
than in peat. Thus, the preferential water flow goes from peat porewaters to pools. The response of the water level to
precipitation was slower and buffered in pools compared to peat (Fig. SI.2) and an average time lag of 13 hours was
measured between the WTD peak of peat and pools.

Peat porewater temperatures were constantly lower than in pools, with 13.4 ± 4.4°C in peat porewaters against 17.1 ±
5.5°C in pools when averaged over the two growing seasons. In both environments, the pH was acidic with an average
of 4.9 ± 0.7 in peat porewaters and 4.4 ± 0.3 in pools. Specific conductivity was on average almost two times higher
in peat porewaters than in pools, with 33.0 ± 19.3 µS cm$^{-1}$ and 14.0 ± 6.1 µS cm$^{-1}$ in peat porewaters and pools,
respectively. Pool waters were characterised by their constant saturation in dissolved oxygen, with 99.9 ± 5.2%sat on
average, while dissolved oxygen saturation was 50.04 ± 17.1%sat in peat porewaters (Table 1).

**Table 1.** Peat porewater and pool seasonal average (±SD) of physicochemical variables (water temperature, pH,
specific conductivity, and dissolved oxygen saturation), dissolved organic carbon (DOC) concentrations, and DOC
to dissolved organic nitrogen (DON) ratio (DOC:DON), isotopic signature of DOC ($\delta^{13}$C-DOC), optical indices
(SUVA$_{254}$, E2:E3 ratio, and spectral slope ratio), fluorescence indices (FI and β:α index), and molecular indices
($f$VEG, $f$MIC deoxyC6:C5, and C/V ratio, %Phenols, %Carbohydrates, %Microbial Carbohydrates,
%SmallOrganicAcids, %LMWFattyAcids, and %HMWFattyAcids).

| | Peat porewater | | | Pools | | |
|---|---|---|---|---|---|---|
| | Spring | Summer | Autumn | Spring | Summer | Autumn |
| **Physicochemical parameters** | | | | | | |
| Water temperature (°C) | 12.3±5.2 | 16.5 ±2.3 | 11.8±4.2 | 14.8±1.7 | 22.8±2.9 | 13.7±4.9 |
| pH | 4.4±0.6 | 5.2±0.3 | 4.9±0.7 | 4.4±0.2 | 4.5±0.2 | 4.3±0.2 |
| Conductivity (µS cm$^{-1}$) | 41.7±23.9 | 26.1±7.4 | 33.5±21.7 | 8.39±1.7 | 12.1±3.5 | 19.1±5.5 |
| Dissolved oxygen (%sat) | 56.0±16.2 | 45.3±16.8 | 50.4±17.5 | 101.0±2.9 | 102.0±5.8 | 97.5±4.9 |
| **Organic matter quantitative proxies** | | | | | | |
| DOC (mg L$^{-1}$) | 9.2±4.2 | 20.2±8.5 | 22.5±5.4 | 7.5±3.2 | 10.4±3.7 | 12.4±4.0 |
| DOC:DON | 32.3±12.4 | 52.8±22.5 | 56.6±8.2 | 26.2±7.7 | 32.0±7.5 | 31.7±6.6 |





| DON (mg L$^{-1}$) | 0.29±0.1 | 0.39±0.08 | 0.39±0.06 | 0.29±0.1 | 0.39±0.11 | 0.39±0.09 |
|---|---|---|---|---|---|---|
| **Isotopic and optical indices** | | | | | | |
| δ$^{13}$C-DOC (‰) | -26.0±0.9 | -27.3±0.3 | -27.5±0.5 | -27.5±0.3 | -27.1±0.4 | -26.8±0.8 |
| SUVA$_{254}$ (L mg$^{-1}$ m$^{-1}$) | 6.0±1.5 | 5.13±0.5 | 5.55±1.0 | 2.88±1.5 | 3.13±0.5 | 3.86±0.6 |
| E2:E3 ratio | 3.4±0.2 | 3.5±0.2 | 3.6±0.2 | 4.0±0.1 | 4.2±0.2 | 4.4±0.2 |
| S$_R$ | 0.67±0.04 | 0.64±0.05 | 0.67±0.03 | 0.77±0.06 | 0.81±0.05 | 0.72±0.05 |
| Fluorescence index | 1.39±0.09 | 1.40±0.13 | 1.33±0.07 | 1.27±0.04 | 1.26±0.03 | 1.27±0.04 |
| β:α Index | 0.63±0.10 | 0.65±0.08 | 0.59±0.05 | 0.62±0.03 | 0.69±0.07 | 0.61±0.07 |
| **Molecular indices and family compound proportions** | | | | | | |
| fVEG (%) | 68.6±1.1 | 69.8±3.6 | 62.7±3.2 | 64.3±10.6 | 60.8±6.7 | 62.8±3.5 |
| fMIC (%) | 5.8±1.4 | 7.2±3.4 | 5.8±2.3 | 8.0±5.2 | 11.7±6.0 | 7.2±2.0 |
| MIC:VEG ratio | 0.09±0.02 | 0.10±0.05 | 0.09±0.4 | 0.14±0.13 | 0.20±0.12 | 0.12±0.04 |
| deoxyC6:C5 | 0.67±0.11 | 0.64±0.25 | 0.50±0.17 | 0.73±0.10 | 1.10±0.19 | 0.91±0.33 |
| C/V | 0.37±0.11 | 0.37±0.13 | 0.22±0.07 | 0.18±0.04 | 0.22±0.08 | 0.19±0.04 |
| %Phenols (%) | 57.6±6.3 | 54.1±9.2 | 53.6±4.6 | 59.0±10.1 | 53.3±8.5 | 54.6±7.9 |
| %SmallOrganicAcids (%) | 19.9±0.4 | 18.1±5.8 | 26.6±6.3 | 21.5±3.9 | 20.4±4.7 | 24.5±3.5 |
| %Carbohydrates (%) | 7.3±1.6 | 5.7±3.2 | 6.7±5.2 | 4.6±2.2 | 8.8±9.4 | 7.8±6.6 |
| %Microbial Carbohydrates (%) | 0.11±0.08 | 0.11±0.15 | 0.05±0.07 | 0.15±0.07 | 0.40±0.13 | 0.28±0.22 |
| %LMWFattyAcids (%) | 5.0±0.7 | 6.7±3.0 | 5.2±1.9 | 7.2±5.9 | 7.6±3.8 | 5.6±1.6 |
| %HMWFattyAcids (%) | 4.6±4.4 | 10.6±8.6 | 3.00±3.3 | 1.4±0.1 | 2.9±1.3 | 2.1±1.0 |

### 4.2.    Evolution of DOC concentrations and DOC:DON ratios

The DOC concentrations in peat porewaters were significantly higher than in pools (Fig.2.a). In both environments, the DOC concentrations showed the same seasonal trends with a significant increase from spring to summer. The DOC concentrations increased significantly in peat porewaters from $9.2 \pm 4.2$ mg L$^{-1}$ in spring, reaching a plateau above 20 mg L$^{-1}$ during summer and autumn. In pools, the DOC concentrations also increased significantly from $7.5 \pm 3.2$ mg L$^{-1}$ in spring to a plateau above 10 mg L$^{-1}$ in summer and autumn.

Peat porewaters presented a significantly higher DOC:DON ratio than pools. In both environments, the DOC:DON ratio increased significantly from spring to a plateau in summer and autumn (Fig. 2.b). In peat porewaters, the DOC:DON ratio increased from $32.3 \pm 12.4$ in spring to $52.8 \pm 22.5$ and $56.6 \pm 8.2$ in summer and autumn, respectively (Fig. 2.b). In pools, the DOC:DON ratio increased from $26.2 \pm 7.7$ in spring to a plateau of $32.0 \pm 7.5$ in summer and $31.7 \pm 6.6$ in autumn.


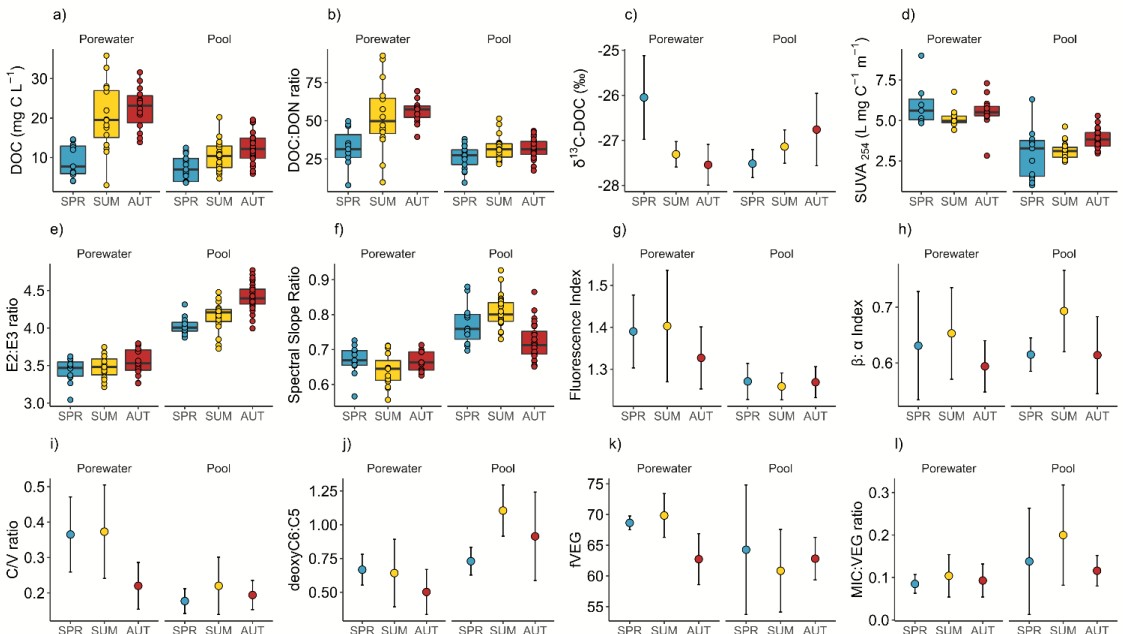

**Figure 2.** Box plots (Figure 2.a to 2.b and 2.d to 2.f) and dot plot (figure 2.c and 2.g to 2.i): a) DOC concentrations, b) DOC:DON ratio, c) $\delta^{13}$C-DOC, d) SUVA$_{254}$, e) E2:E3 ratio, f) spectral slope ratio, g) fluorescence index, h) β:α index, i) *f*VEG, j) deoxyC6:C5, k) C/V ratio, and l) MIC:VEG ratio. Each plot represents the evolution of variables during the growing season (SPR = spring, SUM = summer, AUT = autumn) in peat porewaters and pools. Dot plots were used when n <5 for at least one season. Error bars represent standard deviations. Box plots were used when n >5 for each season. The dots represent each individual measurement, and boxes represent the lower (25$^{th}$ percentile) and the upper quartile (75$^{th}$ percentile); the median (50$^{th}$ percentile) is represented by the bold black horizontal bar in the boxes. Whiskers represent the interquartile range.

### 4.3. Evolution of the isotopic compositions of DOM

Different trends for $\delta^{13}$C-DOC were identified between peat porewaters and pools (Fig. 2.c). In peat porewaters, $\delta^{13}$C-DOC decreased significantly from spring, when the ratio was -26.0 ± 0.9‰, to autumn when the ratio dropped to -27.5 ± 0.5‰. In pools, $\delta^{13}$C-DOC showed a nonsignificant increase from -27.5 ± 0.3‰ in spring to -26.8 ± 0.8‰ in autumn. In summer, $\delta^{13}$C-DOC was significantly different between peat porewaters and pools. When considering all samples, a significant negative correlation was observed between $\delta^{13}$C-DOC and the DOC:DON ratio, with the lower DOC stable isotopic signature corresponding to a higher DOC:DON ratio (cor = -0.53, p-value = 0.0004, Fig. 3).

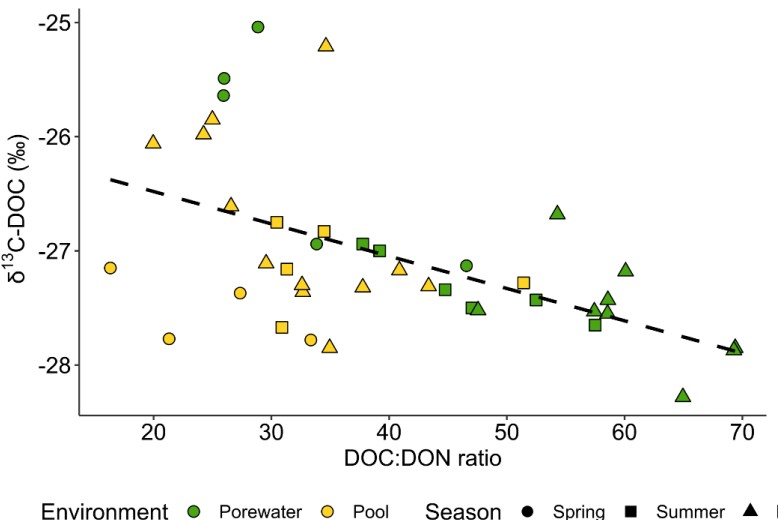

**Figure 3.** Relations between $\delta^{13}$C-DOC and the DOC:DON ratio between seasons in peat porewaters (green) and pools (yellow).

### 4.4. Evolution of the optical and fluorescent properties of DOM

The DOM presented different optical properties between peat porewaters and pools. Among these, SUVA$_{254}$ was

significantly higher in porewaters than in pools during the whole growing season, indicating a higher aromaticity of peat porewater DOM (Table 1). During the growing season, there were no major changes of SUVA$_{254}$ in peat porewaters but a slight increase was observed in pools during the autumn season (Fig. 2.d).

The E2:E3 ratio was significantly higher in pools than in peat porewaters, indicative of a lower average molecular weight in pools. As for SUVA$_{254}$, the E2:E3 ratio showed no significant trends in peat porewaters, but it slightly

increased in pools from $4.02 \pm 0.11$ in spring to $4.41 \pm 0.18$ in autumn, suggesting a decrease of the average molecular weight during the growing season (Fig. 2.e).

The lower spectral slope ratio ($S_R$) of peat porewater DOM also suggested a higher molecular weight than in pool DOM. During the growing season, the $S_R$ was steady in peat porewaters with no significant changes between seasons, suggesting a homogeneity of the molecular weight of DOM (Fig. 2.f). In pools, $S_R$ values increased from spring to

summer and decreased in autumn. Thus, according to the $S_R$, the lowest average molecular weight was reached during summer in pools.

The fluorescence index (FI) was significantly higher in peat porewaters than in pools but varied within a narrow range (Fig. 2.g), close to typical terrestrial-derived organic matter. During the growing season, the index remained steady in both environments with an average of $1.36 \pm 0.10$ in peat porewaters against $1.27 \pm 0.04$ in pools.

The β:α index did not differ significantly between peat porewaters and pools, where it was on average $0.62 \pm 0.07$ and $0.64 \pm 0.07$, respectively (Fig. 2.h). During the growing season, the index remained steady in peat porewaters. In



pools, the β:α index increased significantly from spring to summer: from $0.62 \pm 0.03$ to reach a peak at $0.69 \pm 0.07$. As for the FI, variations of the β:α index were limited to a small range.

### 4.5. Evolution of the molecular composition of DOM

Phenol markers dominated (54%) the molecular markers for both peat porewater and pool DOM (Table 1). A similar proportion of small organic acids (22% on average) was measured in both environments. Carbohydrates represented 6% of the total markers in peat porewaters and up to 8% in pools. The distribution of fatty acids differed between the two environments. While low molecular weight fatty acids showed similar proportions in peat porewaters (5.8%) and in pools (6.7%), high molecular weight fatty acids, which are associated with plant inputs, were almost three times

higher in peat porewaters (6.1%) than in pools (2.3%). This represents the only significant difference in molecular class distribution between peat porewaters and pools (p-value = 0.02, Student test).

Although the distribution of these five classes of compounds remained the same in peat and pools, three modifications of the molecular composition of DOM between these two environments must be highlighted. First, the C/V ratio (Fig. 2.i), a lignin compositional proxy, was significantly higher in peat porewaters than in pools (p-value <0.01). While it

remained almost stable in pools, it decreased in peat porewater sfrom $0.37 \pm 0.12$ during spring and summer to $0.22 \pm 0.07$ during autumn. Secondly the deoxyC6:C5 ratio (Fig. 2.j), a carbohydrate ratio, was significantly higher in pools ($0.97 \pm 0.28$) than in peat porewaters ($0.57 \pm 0.20$) (p-value <0.0001). While it remained almost stable in peat porewaters, in pools, it was maximal in summer ($1.10 \pm 0.19$) compared to spring ($0.73 \pm 0.10$) and autumn ($0.91 \pm 0.33$). This evolution emphasised an increase in the contribution of microbial exudates among the carbohydrate pool in pools. Finally, the fraction of plant-derived compounds among the identified markers, fVEG (Fig. 2.l), was always

higher than 50% in both environments, highlighting the dominance of plant-derived DOM. However, fVEG was significantly higher in peat porewaters than in pools (p-value = 0.02). As for the C/V ratio, fVEG remained almost stable in pools, while it decreased in peat porewaters in autumn.

### 4.6. Global assessment of DOM quality in peat porewaters and pools

The PCA analyses of the peat porewater and pool samples indicate that the two first components, represented by the two axes of Figure 4 accounted for 56.3% of the total variance. Individuals, represented on the two first dimensions, showed a clear separation of both environments along the first dimension (Fig. 4). The major contributors of the first axis were $S_R$ (19.8%), E2:E3 ratio (14.4%), deoxyC6:C5 (12.8%), DOC concentration (12.7%), and finally MIC:VEG ratio (11.8%). For the second axis, the major contributors were the proportion of phenols (%PHENOLS; 20.8%), C/V

ratio (18.5%), high molecular weight fatty acids (%HMW FA; 17.3%), and DOC:Cl ratio (10.4%). Other variables contributed less than 10% to the two first axes.

In pools, the DOM was characterised by a lower average molecular weight and aromaticity and a higher contribution of microbial-derived DOM compared to peat porewaters. Inversely, in peat porewaters, the DOC concentrations were higher and DOM presented a higher aromaticity, and a higher contribution of plant-derived DOM, characterised by a

higher fVEG. There was no effect of the sampling season on the variances.



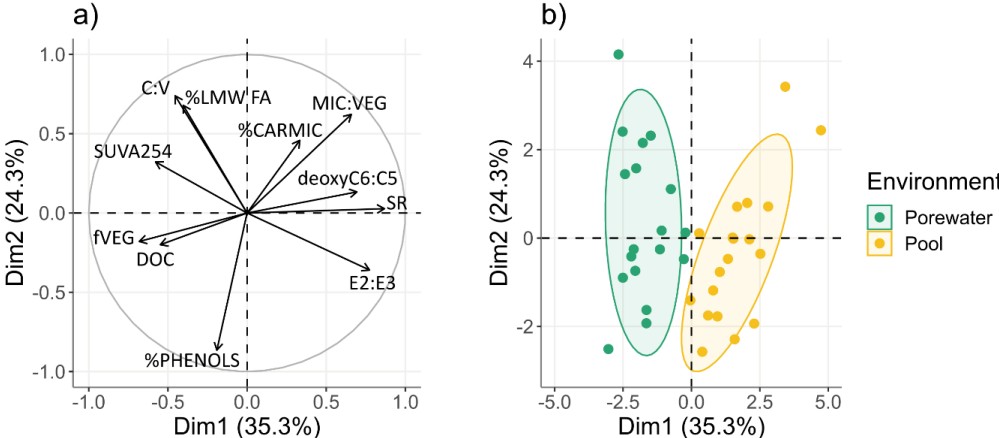

**Figure 4.** Representation of the first two dimensions of principal component analysis (PCA) of a) individuals and b) physicochemical, quantitative, and qualitative parameters as variables. *DOC* corresponds to DOC concentrations and represents the only quantitative variable. The qualitative variables are *DOC:Cl* for the DOC:Cl ratio, *SUVA254* for the absorbance index SUVA$_{254}$, $S_R$ for the spectral slope ratio, and *E2E3* for the E2:E3 ratio. The variables *%PHENOLS, %HMW_FA* correspond to the proportions of phenol makers and high molecular weight fatty acids, respectively. The molecular indices are *MIC:VEG* for the microbial against plant-derived compound ratio, *deoxyC6:C5* for the deoxyC6:C5 ratio, and *C:V* for the C:V ratio. The two first dimensions explained 56.6% of the total variance. The ellipses correspond to the function *addEllipses* from the R package FactoMineR used to add concentration ellipses to the plot.

### 4.7.    Experimental degradability of peat porewaters and pool DOM

Statistical tests revealed no significant differences in the degradation rates between *in situ* and controlled conditions of biodegradation (section 3.5.1). In addition, no significant differences appeared between the degradation rates where biodegradation only was tested and those where biodegradation and photodegradation were both tested. This suggests that temperature and sunlight had a limited effect on the DOM degradation. As a consequence, all experimental conditions (both *in situ* and controlled) are pooled in the following section. The DOM degradation rates were significantly higher for peat porewaters than pools. Statistical tests also revealed that degradation rates were significantly higher for the incubation conditions of unfiltered samples (UF) compared to filtered sample (F) conditions (Fig. 5). On average, the DOC degradation rates were 1.6 times higher for incubation under unfiltered conditions (2.5 ± 1.5%C day$^{-1}$) compared to filtered conditions (1.5 ± 0.8%C day$^{-1}$) in peat porewaters. In pools, degradation rates were twice as high for UF (1.1 ± 1.1%C day$^{-1}$) than for F conditions (0.5 ± 0.6%C day$^{-1}$).

In peat porewaters, the DOC degradation rates for F and UF conditions followed similar seasonal trends. The DOC degradation rates were low in June (0.6 ± 0.4%C day$^{-1}$) and twice as high for UF conditions (1.3 ± 1.0%C day$^{-1}$). The degradation rates reached a peak in August, with 2.2 ± 0.5%C day$^{-1}$ for F and 4.5 ± 0.8%C day$^{-1}$ for UF conditions.





Then, the DOC degradation rates decreased in autumn to $1.7 \pm 0.6$ %C day$^{-1}$ and $2.2 \pm 0.5$ %C day$^{-1}$ for F and UF incubation conditions, respectively.

There is no value available for F conditions in pools in August due to some technical problems. In June, the DOC degradation rates were similar between the F and UF conditions with rates of $1.0 \pm 0.5$ %C day$^{-1}$ and $1.3 \pm 0.2$ %C day$^{-1}$, respectively. Then, an increase to $2.1 \pm 1.3$ %C day$^{-1}$ was observed in August for UF conditions which was two times

lower than the rate measured in peat porewaters under the same conditions. Finally, the DOC degradation rates diminished in September, with $0.8 \pm 0.3$ %C day$^{-1}$ for F and $1.1 \pm 0.3$ %C day$^{-1}$ for UF incubation conditions. Those degradation rates were two times lower than those observed in peat porewaters in autumn.

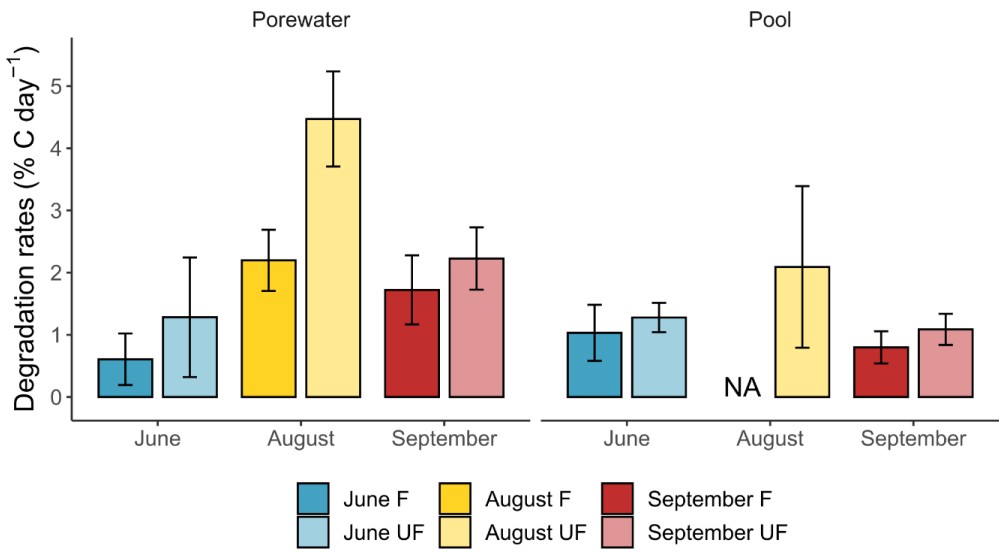

**Figure 5.** Seasonal degradation rates (in %C day$^{-1}$) for DOC and TOC incubation conditions in peat porewaters and pools.

Changes in SUVA$_{254}$ ($\Delta$SUVA$_{254}$) during incubation experiments were linked with degradation rates. Under most of the incubation conditions, SUVA$_{254}$ increased. As observed in Figure 6, $\Delta$SUVA$_{254}$ was strongly and positively

correlated with degradation rates in pools (n = 29, cor = 0.82; p-value <0.0001) and peat porewaters (n = 33, cor = 0.65; p-value <0.0001).



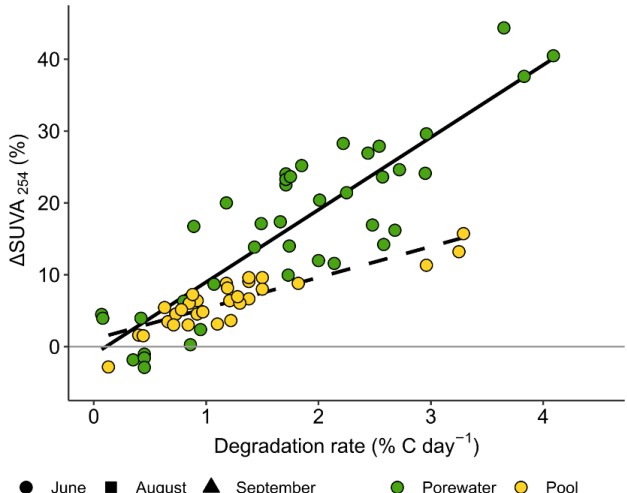

**Figure 6.** Relations between changes in $SUVA_{254}$ ($\Delta SUVA_{254}$) during incubation experiments and linear regression in peat porewaters (dashed line) and pools (solid line).

## 5. Discussion

### 5.1. Differences in DOM concentrations and composition between peat porewaters and pools hide similar sources

The DOC concentrations measured at our sites (Table SI.3) are in agreement with the expected range of a subarctic peatland (Deshpande et al., 2016). A synthesis of the DOC concentrations in peat porewaters highlights a latitudinal gradient of DOC concentrations in peatlands, from DOC concentrations commonly lower than 20 mg $L^{-1}$ in boreal and sub arctic zones compared to temperate zones, suggesting a temperature control on the balance between DOM production and processing (Kane et al., 2014).

The peat porewater DOM presented specific features, including high values of average molecular weight, aromaticity, and DOC:DON ratio (Fig. 2 and 4). The $SUVA_{254}$ value of 5.5 L $mg^{-1}$ $m^{-1}$ measured in the Bouleau peatland was, in general, higher than those previously measured in temperate peatlands, i.e., <3.6 L $mg^{-1}$ $m^{-1}$ (Arsenault et al., 2019; Tfaily et al., 2015; Heinz and Zak, 2018), except for Austnes et al. (2010) who reported a similar aromaticity and average DOM molecular weight as in the Bouleau peatland. It is more difficult to assess any latitudinal gradients for DOM compositions, since fewer studies are available. The DOC:DON ratios measured in peat porewaters at our study site were up to six times higher than in Austnes et al. (2010). These high ratios, as well as the negative correlation observed between the DOC:DON ratio and $\delta^{13}$C-DOC (Fig. 3) reflect the contribution of plant leachates as opposed to microbial exudates (Magill and Aber, 2000).

The measured pool DOM also presented specific features, with lower DOC concentrations, aromaticity, and average molecular weight compared to peat porewater DOM (Fig. 2). The DOC concentrations in pools were similar to those



previously reported in the literature (Table SI.3) which, unlike peat porewaters, do not present any latitudinal distribution. The SUVA$_{254}$ values measured in the Bouleau peatland pools (3.4 ± 0.9 L mg$^{-1}$ m$^{-1}$ on average) were similar to those reported from Arctic regions with values of about 4 L mg$^{-1}$ m$^{-1}$ (Laurion and Mladenov, 2013; Peura et al., 2016; Gandois et al., 2019; Laurion et al., 2021) suggesting a terrestrial contribution at our site.

Despite the contrasting concentrations and compositions between peat porewaters and pools, our results indicate a similar dominant vegetation origin of DOM in both environments. The molecular compositions of the peat porewaters and pools highlighted the dominance of phenol markers (>50% of total markers) related to plant sources. The molecular indices $f$VEG and deoxyC6:C5 also supported the dominance of a plant origin of DOM in both environments (Fig. 2). The stable isotopic signature of DOC from peat porewaters of -27.0 ± 0.8‰ on average varied

within the same range as that observed for DOC derived from peat porewaters and peat, -27.9 and -27.8‰, respectively (Elder et al., 2000; Clymo and Bryant, 2008; Buzek et al., 2019).

The decrease of the DOC stable isotopic signature and increased DOC:DON ratio in the peat porewaters, as well as the contribution of high molecular weight fatty acids from spring to summer highlighted a greater proportion of plant-derived DOM towards the end of the growing season. At the same time, the low dissolved oxygen saturation could

reflect the bacterial consumption of oxygen (Table 1). This, coupled with the presence of microbial markers expressed by $f$MIC (Table 1) and the evidence of a labile fraction of DOM (Fig. 5) underline the importance of the microbial processing of DOM within peat porewaters. All this highlights that both DOM production and DOM microbial processing are active within peat porewaters, with DOM production being more intense, as DOC concentrations were multiplied by 2.5 during the growing season (Table 1).

Our data did not evidence any photodegradation during DOM incubation in peat porewaters and pools, suggesting that the DOM was potentially not photolabile. This contrasts with previous studies which observed DOM photodegradation and changes in DOM composition in boreal and temperate aquatic ecosystems of Eastern Canada (Lapierre and del Giorgio, 2014; Ward and Cory, 2016) and the United Kingdom (Jones et al., 2016). Because degradation processes can occur within the first days following DOM production (Vonk et al., 2015), the already degraded DOM we sampled

might not have presented any sizeable photolability (Cory and Kling, 2018; Shirokova et al., 2019). The clear pattern of the SUVA$_{254}$ increase observed during the incubation period was independent of the conditions where DOM was exposed to solar radiation. This highlights the preferential consumption of non-aromatic molecules (Spencer et al., 2008, 2015; Mann et al., 2015; Worrall et al., 2017), leading to an increase of SUVA$_{254}$ (Hulatt et al., 2014; Autio et al., 2016) while photooxidation has been shown to induce a decrease of DOM aromaticity (Laurion and Mladenov,

2013; Ward and Cory, 2016).

### 5.2. How hydrological, chemical, and biological processes explain the compositional differences of DOM between peat porewaters and pools

The observed differences in DOM compositions between peat porewaters and pools despite similar dominant plant-derived sources, can be explained by a combination of hydrological, chemical, and biological factors.

Hydrological pathways at the interface between peat and pools might play a role in the shift of DOC concentrations and DOM composition between porewaters and pools. The two environments appear to be hydrologically connected, based on the synchronous variations of the water levels in adjacent environments with strong buffering in pools (Fig.

SI.2). The buffering of water level variations in pools can be explained by the decrease of hydraulic conductivity with depth in peat which limits water exchanges (Holden et al., 2018).

In the Bouleau peatland, a decrease with depth in the water storage coefficient (Riahi, 2021) and an increase in peat density was observed (Primeau and Garneau, 2021), limiting water circulation. This slower circulation of water induces a longer residence time of DOM with peat depth and favours interactions with microorganisms, allowing for the microbial degradation of DOM (Kalbitz et al., 2000). Evidence of microbial degradation in the peat was suggested by the high SUVA$_{254}$ as expressed by the characteristics of degraded DOM observed through degradation experiments

(Fig. 5) and the observation of microbial markers (e.g., $f$MIC, Table 1)

The structure of peat pores also stimulates interactions between DOM and partially degraded peat which can adsorb both hydrophilic and hydrophobic compounds (Kalbitz et al., 2000; Rezanezhad et al., 2016). Changes in composition between peat porewaters and pools might be induced by the selective interaction of DOM compounds in peat during their slow transfer. Since aromatic compounds are known to constitute the hydrophilic fraction of DOM (Dilling and

Kaiser, 2002) this process might explain the lower aromaticity measured in pools (Table 1).

Both the concentrations and compositions were observed to change sharply at the interface between the peat and an adjacent pool (Fig. SI.4). This highlights that the shift in DOM composition and concentration was not progressive through its transfer through peat but might have been enhanced at the interface between those environments. In pools, the higher contribution of microbial-derived DOM, expressed by higher deoxyC6:C5 and lower $f$VEG indices, and

the decrease of the average DOM molecular weight shown by the higher $S_R$ and E2:E3 ratio, support the hypothesis of a reactive interface (Fig. SI.4). A shift in physicochemical parameters between the two environments, such as the slight increase in pH and temperature, and the rise of dissolved oxygen concentrations, may favour the microbial turnover of a fraction of DOM exported to pools (Schindler et al., 1997; Kalbitz et al., 2000; Worrall et al., 2008; Peura et al., 2016). The higher potential degradation rate that was measured for peat porewater DOM suggests that a

fraction of DOM which could be transferred to pools is labile and can actually be degraded at the interface between peat and pools. In addition to the DOM processing at the interface between peat porewaters and pools, the molecular composition of pool DOM reflected microbial activity within the pool itself. The relative proportions of these processes (DOM transformation within the peat, at the interface, or within the pool) cannot be assessed.

### 5.3. Implications of the DOM of pools in the peatland carbon budget

Boreal peatland pools were previously identified as a continuous source of $CO_2$ to the atmosphere during ice-free seasons, offsetting some of the carbon uptake by the vegetation (Pelletier et al., 2014, 2015). Therefore, the microbial processing of DOM in pools is likely to generate carbon dioxide emissions (Billett et al., 2004; Striegl et al., 2012; Payandi-Rolland et al., 2020). Since our results showed low apparent degradation rates in pools, we suggest that DOM could have been partially biodegraded within the peat and at the interface between peat porewaters and pools, limiting

further its degradation within pools (Payandi-Rolland et al., 2020).

Incubations of DOM that included particulate organic carbon (POC) under unfiltered conditions revealed higher degradation rates than those measured for filtered conditions, indicating that POC might stimulate the degradation of DOC (Fig. 5). Indeed, POC was previously described as a hotspot of biodegradation because particulate compounds



stimulate interactions between bacteria and organic substrates (Attermeyer et al., 2018). Degradation rates under
unfiltered conditions in pools were two times lower than for peat porewaters. This might be related to the very low
levels of turbidity in pools (data not shown). This is consistent with the decrease of DOM degradability previously
observed at the transition from peat to aquatic environments (Hutchins et al., 2017; Payandi-Rolland et al., 2020).

The long-term apparent rate of carbon accumulation (LORCA) measured in the Bouleau peatland has been estimated
to be 35.5 g C m$^{-2}$ yr$^{-1}$ and the recent apparent rate of carbon accumulation (RERCA) 85.1 g C m$^{-2}$ y$^{-1}$ (Primeau and
Garneau, 2021). Based on a total pool volume of approximately 136 350 m$^3$, an average DOC concentration of 10.1
mg L$^{-1}$ (Table 1), and an average potential degradation rate between 1.9 % day$^{-1}$ in peat porewaters and 0.9 % day$^{-1}$ in
pools (Fig. 5), the degradation of pool DOM could average between 1.5 and 3.1 g C m$^{-2}$ yr$^{-1}$ for our site. This is
equivalent to 4.2 to 8.7% of the LORCA and 1.7 to 3.6% of the RERCA, supporting the relatively slight effect of the
processing of DOM in pools on the peatland carbon budget. However, DOM degradation is not the only source of
carbon emissions from pools, which can also be supplied by the lateral transfer of $CO_2$ and $CH_4$ (Rasilo et al., 2017)
and by $CH_4$ ebullition (Repo et al., 2007).

The studied site is an undisturbed boreal peatland, without permafrost or direct human pressures that could influence
the carbon production and transformation processes through the peat-pools complex. The morphology of pool banks
and vegetation surrounding the pools may play an important role in the DOM dynamics and DOC concentrations of
pools, as suggested by Arsenault et al. (2018, 2019). A study conducted on ten peatland pools showed that the size of
the contact surface between water and peat (influenced by pool size, depth, and the slope of the banks) influenced the
concentrations and composition of DOC (Banaś, 2013). Even if the DOM of pools appeared to have a limited effect
on the net carbon budget of our studied peatland since pool banks were sharp and surrounded by upland vegetation,
further studies should be conducted within different peatland ecosystems to better document the spatiotemporal
variability and fate of DOM under contrasting climatic regions of the northern hemisphere.

## 6. Conclusions

This study highlighted that DOM is a highly dynamic component of the carbon cycle in boreal peatlands, with
important changes in its concentrations and composition in both peat porewaters and pools during the growing season.
The strong increase of DOC concentrations in peat porewaters over the growing season highlighted the intense
production of DOM in peat porewaters. Indeed, DOC concentrations increased by 2.5 during the growing season, even
with the microbial processing of DOM occurring in peat. The molecular analysis of DOM revealed that peat porewater
and pool DOM shared the same dominant vegetation origin, despite exhibiting very different concentrations,
compositions, and dynamics over the growing season. Based on our investigations, we suggest that a combination of
hydrological, chemical, and biological processes explains these differences.

The low hydraulic conductivity in peat might favour the contact of DOM with microorganisms resulting in its
degradation before its transfer to aquatic compartments. Low hydraulic conductivity might also lead to the selective
adsorption of aromatic compounds with degraded peat supporting the decrease of concentrations and the lower
aromaticity of DOM observed in pools. We observed abrupt changes in DOM concentrations and compositions at the
interface between peat and pools. The rapid modification of physicochemical parameters between those environments



might favour the biodegradation of DOM at the interface between the peat and the pools and within the pool. This is confirmed by the higher proportion of microbial molecular markers identified in the pool DOM. Although DOM is microbially processed both at the interface and within the pool, the carbon emissions generated by these processes remain small in comparison with the carbon accumulation potential of the peatland.

**Data availability**

**Author contribution**

Conceptualisation: Laure Gandois, Michelle Garneau, Antonin Prijac

Data curation: Laurent Jeanneau, Antonin Prijac, Pierre Taillardat

Data analyses: Laurent Jeanneau, Antonin Prijac

Formal analyses: Antonin Prijac, Laure Gandois

Funding acquisition: Michelle Garneau

Investigation: Laure Gandois, Antonin Prijac

Methodology: Laure Gandois, Laurent Jeanneau, Antonin Prijac

Data collection: Antonin Prijac, Pierre Taillardat

Writing – original draft preparation: Antonin Prijac

Writing – review and editing: Laure Gandois, Michelle Garneau, Laurent Jeanneau, Antonin Prijac, Pierre Taillardat

**Acknowledgments**

The funding for this research was provided by the Natural Sciences and Engineering Research Council of Canada and Hydro-Quebec to Michelle Garneau (RDCPJ 514218-17). We thank Katherine Velghe and Alice Parks, from GRIL, for their laboratory training and assistance in absorbance and fluorescence analyses, and Paul Del Giogio for access

to his laboratory. Marine Liotaud, from Geosciences Rennes, is acknowledged for performing TMH-GC-MS analyses and Frederic Julien, Virginie Payre-Suc, and Didier Lambrigot, from Laboratoire Ecologie Fonctionnelle et Environnement, for performing DOC/TN and cations/anions analyses. Thanks to Charles Bonneau, Charles-Élie Dubé-Poirier, Camille Girard, Pénélope Germain-Chartrand, Léonie Perrier, Guillaume Primeau, Khawla Riahi, Roman Teisserenc, and Karelle Trottier for their assistance in the field.

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
