# Peer review of "Dissolved organic matter concentration and composition discontinuity at the peat-pool interface in a boreal peatland"

_Biogeosciences, 2022_

## Referee Comment (RC1)

**Prijac A. 2022 . Biogeosciences**

**Overview:**

This paper presents a large and detailed investigation of DOC concentration and composition in peat porewater and pools in a pristine peatland in northern Quebec. The dataset is interesting. As clearly stated in the title, the study reveals major "discontinuities" in DOC concentration and composition between porewater and pools within a peatland. This could have been done with a single spatially distributed sampling, but the authors complement this dataset by repeating the sampling over different seasons. The seasonal sampling corroborates the initial findings of "discontinuity" in the DOC concentration and composition between these two environments. Whatever hydroclimatic conditions, the DOC in the pools and surrounding peat porewater seems to be considerably different. Overall, I find the dataset presented here to be interesting and the methods and statistics are sound. However, I have some concerns over the interpretation of the findings and how they support the conclusions of the study. In addition, I have made some recommendations to improve the data visualizations and some elements of the text, mainly the discussion.

**Causes of Discontinuity:**

The peat reaches 4m deep in some locations (line 106) (often near the pools based on the map in Pimeau and Garneau 2021), but the porewater sampling considered only the top 2m (Line 125). I expect the reason for that is the assumption that hydraulic conductivity decreases exponentially with depth (stated in discussion 460-466). Therefore, the porewater in the bottom 2m of the peat profile is considered to move very slowly and contribute little to runoff generation or the water contained in pools.

However, deep preferential flow areas exist in many peatlands (often below 2m deep) (e.g. two Swedish studies DOI: 10.1111/gcb.13815, DOI: 10.1002/hyp.10300 (one where I was involved, sorry for citing myself), UK peatlands (e.g. DOI:10.1016/S0341-8162(01)00189-8) and GLAP peatlands (DOI:10.1002/2016GB005397, DOI: 10.1002/hyp.9983 ). These studies have shown that deep peat horizons can contribute to a large fraction of the runoff generation, or at least be hydrologically active. The high hydrostatic pressure in deep peat preferential flow areas can make that water emerge rapidly to the surface in specific locations, for example in streams or pools. Could this also be the explanation here? The water and C found in pools could in fact be feed predominantly from the bottom instead of laterally? This could also explain why the water table is more stable in the pools than the peat porewater (Line 300, Fig SI2). Hence, the discontinuity in DOC concentration and composition observed here could in fact be due to that the sampling didn't capture the actual source of water and C (if it is located below 2m deep). The difference in specific conductivity between the pools and peat porewater, which here could act as an independent water tracer, indicates different water sources (Line 304-306).

Assuming the authors finds this to be a plausible explanation, I want to highlight that I still find the results of the study to be interesting and relevant by simply highlighting the major and persistent disconnect between surface porewater and pool water. This disconnect possibly arises as a result of the complex hydrology of peatlands. Maybe the paper would benefit from emphasizing this disconnect rather than to suggest a "common source" and find a reason for the apparent discontinuity (which are suggested to be a combined result of hydrological, chemical and biological process) (see my comment on the discussion). Maybe it's my background showing here, but I believe a missing water source could explain nearly all the observed patterns in this dataset.

**The Common Source:**
The section 5.1 of the discussion claims that the differences in DOC concentration and composition hide a common source. The author claims that common source to be C3 plant-derived, which leads me to wonder - what else could it have been in a boreal peatland? I have several issues with this aspect. 1. It seems obvious to me that the DOC would be predominantly plant-derived in a peatland and therefore doesn't constitute a hypothesis that needs testing nor a substantial finding, 2. The author hasn't clearly stated what other possible sources could be, I suppose these are "*C4 plants*", which generally are absent at this latitude or "*microbial-derived*" which are certainly overridden by the decomposing peat material. 3. The vocabulary often changes, sometimes this source is referred to as "terrestrial contribution" "vegetation origin" "plant-derived". I suggest that the other clarifies both the hypothesis tested here and the vocabulary.

This leads me to further concerns over the interpretation of the d13C-DOC values and the correlation between d13C-DOC and DOC_DON ratio in Figure 3. The d13C-DOC values reported here varies across a narrow range (-25 to 28‰) and show that DOC originated from C3 plant metabolism. Meanwhile the C:N ratio show that the DOC is strongly terrestrial as opposed to aquatic. This is not surprising. But I doubt that anything else can be said of these values. What is the interpretation of the correlation Fig 3? the discussion mentions this correlation briefly on line 421-423 and Line 434-436 (albeit a missing reference to the figure here). The authors state that this correlation reveals that DOC comes from plant leachates instead of microbial exudates, but I don't see how this is supported. The study cited here (Magill and Aber, 2000) has no mention of the stable C isotopes. Are you suggesting that there is a fractionation process taking place here, whereby DOC becomes lighter with increasing DOC:DON ratio due to microbial or photodegradation? Below is a typical biplot from a review paper on d13C-DOC and DOC:DON across ecosystem types that puts in context your data (https://doi.org/10.1016/j.earscirev.2005.10.003).

[Figure]

It seems that the correlation between d$^{13}$C-DOC and C:N ratio might be just coincidental. The mechanism to explain why these two variables should be correlated has not be stated. Further clarification of the interpretation of this correlation would be helpful.

**The role of photodegradation in peatlands (Line 445 to 455)**
I find interesting that the authors quantify the potential photodegradability of the DOC to understand the possible fate of the DOC in downstream environments. But I doubt that photodegradation within this peatland catchment can possibly be an important process for the overall peatland C budget, given
1. the limited amount of light penetrating in peat and pools, and 2. The small areas covered by the
pools. The DOC being transported with water will eventually leave the catchment boundary and maybe
then photodegradation can then play a role. I find interesting that this aspect was quantified as a way
to characterize DOC properties and act as another tracer of DOC sources, but the way it's presented
here, in a context of a mass budget and as a possible mechanism for the disconnect in DOC between
porewater and pool seems overstretched.
**The influence of DOC adsorption**. The peat porewater samples were collected through a PVC tube
covered with a nylon sock. I would assume that the porewater DOC that is adsorbed to the peat to not
sampled then. So can this mechanism really be important to explain the discontinuity between pool
and porewater DOC?
**Influence of Bioavailability (Line 441 to 444):** Are local differences in DOC lability really important for
a peatland given that other environmental factors limit the metabolism in peat porewater. It's again
interesting to measure the lability of DOC as a tracer of DOC sources, but other possibly more
important factors limit the degradation of DOC in peat soils. The author also states that the slow
hydraulic conductivity increases residence time and therefore the potential transformation of DOC.
This is true for most surface water environment (e.g. 10.1038/ngeo2720), but other possibly more
important factor (e.g. electron acceptor availability) limit microbial metabolism in peat. I am not sure
this is a relevant argument here, especially since hydraulic conductivity and water residence time were
not quantified here as far as I am aware. DOC will be degraded once the environmental conditions
allow it, and that is possibly outside of the peatland catchment boundary.
In general, I find that the text in the **discussion** could be improved. I have the feeling that the author is
not fully satisfied with the main conclusion of the paper and tries to find more "positive" results to give
value to the study. In my opinion, the fact that there is such an obvious mismatch between the DOC
concentration and composition across seasons is not a failure, but an opportunity to reveal how
dynamic peatland hydrology and DOC cycling can be. I would suggest that the author rework the
discussion to emphases those differences throughout the discussion, instead of suggesting that there
is a "hidden common source" (which is would always be plant-derived) and that those differences
could be the result of hydrological, chemical, and biological processes. Why not embrace the idea that
the source of DOC in surface porewater and pools are not the same (i.e. section 5.1), and discuss the
possible reasons (section 5.2) and what the implications might be for role of pools in the DOC cycling
of this peatland (i.e. section 5.3). What I am recommending here is not a major reworking of the
discussion, but just a change in perspective, which could make the text more dynamic and with a
clearer message. The title of the paper clearly highlights the main finding: " there is discontinuity in the
DOC between peat porewater and pool across seasons" but the text seems to try to argument that this
DOC might in fact be the same, it's just been biologically transformed/adsorbed to peat etc.  I have
made more specific suggestions on the subject in the "by line comment" section.
**Data Visualization:**
Figure 2 is a key figure presenting the data, but I find the box plot to be an ineffective choice of
visualization in this case. There are too many plots and too many things being compared for this type
of plot to work. A more effective visualization could be a parallel coordinate plot (e.g.
https://datavizcatalogue.com/methods/parallel_coordinates.html). Each vertical axis would represent
a different variable (e.g. DOC concentration, DOC:DON etc.), and each line moving laterally would be a
different sample location. I would suggest to fade the peat porewater samples in the background and
superimpose the pool water samples on top in a darker color to help compare these two environments. You could even make it a three panel figure, one for each season. This would allow to
see at a glance, which site/season bear most similarities and differences for all variables. If possible,
maybe also indicate the meaning of optical properties on the axis, for example (higher SUVA values is
more aromatic and lower is less aromatic etc.). This would facilitate the visual interpretation. This plot
would also give the possibility to merge figure 5 and figure 2, in this case by adding another vertical
axis for degradation rates.
If you choose to stick to the boxplot format please add letters on the x-axis to show statistically
different groups. (e.g. function multcompLetters in R, library multcompView)
Can the symbols in the PCA (Figure 4) be the same at in Figure 6? Also note that all symbols in figure 6
are circles so there is an error here. The figure 4 could also be bigger for better readability.
Figure 1: Is it relevant to point the stream outlet?
**By line comments:**

**Line 69 to 78:** Those lines might fit better in the discussion if you choose to emphasize the differences
in DOC sources between porewater and pools.
**Line 383:** Is it the "average" or "median" degradation rate that was statistically significantly different?
**Line 410:** Be more specific here. Is it the average or range in DOC, porewater or pool water? It can also
be helpful for the reader that you write in bracket the number you are referring to, even if they are
available in the table SI3.
**Line 412-414**: Do you mean here that the subarctic and boreal peatlands have on average 20 mgCL less
DOC than temperate ones? Be more clear about what you are comparing here. Also, sentences starting
with " A synthesis … " or " a study …"  make the text more tedious. You can go straight to the point
here and say for example. Porewater DOC concentration in peatlands exhibit a strong latitudinal trend,
whereby boreal and subarctic peatlands contain …". Also, its not clear why this study is mentioned
here and what argument you are trying to make. Is this just to state that your DOC concentration are
"normal" or are you trying to extrapolate your findings to other latitudes
**Line 420**: same comment as for the previous paragraph: why is it relevant to put in context your
number with latitudinal trends? Be more explicit about what you are trying to say here
**Line 421:** Instead of comparing with the name of the study, you could refer to the type of peatland
that was studied in this paper Also, why are you making this comparison, again the argument is
missing here: The DOC:DON ratios measured in peat porewaters at our study site were up to six times
higher than in Austnes et al. (2010) " *suggesting that* …".
**Line 452:** This is a hypothesis here, no? The photodegradable fraction of DOC might have already been
degraded prior to sampling, but the way it's written here makes it sound like you are certain that's the
case.
**Line 453**: By consumption here you are referring to the biological pathway, not the photochemical
one. Please clearify.

**Line 462**: can they "be explained by" or they can "arise as a result of". This sounds like you are
pleading a case for more "positive" result, while a more "negative" result in this case can be even
more interesting.
**Line 463**: That doesn't mean that water cannot be constantly filled from the bottom and just
occasionally sourced from surface peat when the water table is high.
**Line 410:** The reference to table SI3 should be placed at the end of the sentence.
**Line 410-415**: Why mention this latitudinal effect? There seems to be an argument missing here. Do
you mean that your data fit in with other peatlands at the same latitude? Please clarify.
**Line 507-517**: If the pools allow old DOC to make its way to the surface and therefore enter the
peatland contemporary C cycle then they become very important for the stability of the old peat C
stock. But based on the molecular weight indexes, the DOC seems younger in the pools than
porewater
**Line 520:** My personal suggestion for future studies would have been to combine studies on DOC
cycling with interdependent water tracer. It's hard to trace back the cycling of DOC without knowing
the water source.
**Line 522**: Exactly! I would even add, that the disconnect between the two environments persist no
matter what the hydroclimatic conditions are.
**Line 525**: Or is the concentration just increasing because it gets drier? Maybe check if this difference
persists once the DOC is volume-weighted based on water table position?
**Line 534:** the term "physicochemical parameters" is vague.
**Line 535:** What you say here is true, but I find it to be a disappointing end to a paper. The last sentence
of the conclusion is very important and I am sure there is more a interesting take home message to be
given here.
**Table SI.3:** DOC mean and SD?
**Fig S1:** If E2:E3 and Sr are both proxies of molecular weight (Line 189), why is it that they correlate so
poorly?
Audrey Campeau

---

## Referee Comment (RC2)

**Referee comments on "Discontinuity of the concentration and composition of dissolved organic matter at the peat-pool interface in a boreal peatland" by Antonin Prijac et al.2022**

**MS No.: bg-2022-71**

**General comments**

This paper examines the difference in DOM concentration and composition between peat soil and pools, and discusses the factors leading to this discontinuity. I mostly enjoyed the paper. The design and analysis are understandable, and various methods were used to provide information on DOC composition and source. Furthermore, peatland management is currently an important topic of research, but the C cycling dynamics in boral peatland pools is still largely unknown. Thus, papers on this topic should be welcomed. However, there are some relatively major problems with the paper as it currently stands, about the methods and discussion. Some mistakes including typos should also be addressed to improve the paper quality.

**Specific comments**

**Materials and methods:**

In general I think the writing of method is a bit lengthy. While it's good to provide such information for readers who want to replicate the method, there are too many details which are not necessary to be included in the main text of the paper. I'd like to suggest the authors to refine this part in a concise manner, combining with references and supplementary materials.

In addition, as various sampling trips, analyse methods, proxy indices are used in this study, I'd like to suggest using a table or diagram to summarise this information, which would make it much easier for the readers to understand the research design and interpretation. For example, how many samples from what sites on which dates, and which were analysed for what. It'd also be helpful for the readers to understand why dot plots were used in Fig. 2.

**Discussion:**

1. **Difference in DOM concentration and composition**

Firstly, when comparing the DOC concentrations in porewater and pools in this study with those in other climatic zones, seasonality should be considered, as here DOC samples were collected in the growing seasons which would tend to be higher than other seasons.

Secondly, the authors only explained the good correlation between DOC:DON and 13C in porewater, but didn't try so with the pool DOC. The absence of this correlation in the pools could well lead to the discussion in 5.2, and highlights the discontinuity of DOC composition.

Lastly, while the photo-degradable DOC might have been quickly degraded in the first few days, I'm not entirely convinced that it would be the main reason. DOC from porewater was not exposed to light before being collected so there should be minimal effects from photodegradation. In pools, new DOC inputs would be expected with the increased precipitation, which was observed especially during the summer and autumn seasons. Therefore, there could be continuous supply of photo-degradable DOC during those periods of time. Furthermore, in boreal and arctic areas, the amount of sunlight is less abundant than low latitude areas, limiting the photodegradation of DOC, although I realised it would be less so in summer. Did you have any data on the light? Was any incubation conducted when it was rainy or cloudy? Did the glasses/vials filter out certain wavelengths which cause photodegradation? As it stands now I don't think there is enough evidence to make the argument that there was no photodegradation process in the samples.

**2. Processes leading to the difference**

In general, I think the authors can refine this section by focusing on the more likely processes with a bit more in-depth discussion. It reads like the authors were only including different reasons (e.g. hydraulic conductivity, structure of peat pores, water storage coefficient) that could possibly happen in pools and porewater, but lacking the strength to combine them together. In addition, a few arguments that the authors made were not clearly explained and need to be clarified.

Firstly, the difference in DOC composition, e.g. $SUVA_{254}$, E2:E3, could be a result of that DOC in peat soil is 'older' than those exported to water, which was not considered in the paper before thinking about the more instant changes caused by hydrological, chemical and biological processes as the authors focused on. Also, the aromatic DOC should be more hydrophobic rather than hydrophilic. If aromatic compounds are hydrophilic, it should be easier for them to flow from peat to pools leading to higher DOC aromaticity in the water, which is contrasting to what was observed. In addition, in the abstract the authors pointed out the transformation of DOC at the interface led to the production of low molecular weight compounds, which is contrasting to their suggestion that microbial-processing would cause the increase in aromatic DOC which is often larger in molecular weight.

Secondly, DOC does interact with different materials or minerals in peat. For example, the oxidation/reduction of Fe have been observed to be mediated by microbes and affect the solubility of DOC (Mladenov et al., 2010), and tend to coagulate with high molar mass DOC (Ritson et al., 2014). At the interface between peat and pools, particularly when water table is higher in peat, the change from anaerobic to aerobic environment could affect the reduction/oxidation of certain relevant minerals (e.g. Fe) and reduce mobility of certain group of DOC (Nierop et al., 2002).

Lastly, it was great to see that the authors were trying to explore the biogeochemical processes from porewater, interface and pools, which could be a highlight of this paper. However I'm not entirely sure how big role microbial processing is at the interface as the authors claimed. Indeed, biodegradation could happen within a couple of days, but at the interface I tend to think the physical and chemical interactions, e.g. precipitation and binding via the changes in the physical environment from soil to water is more instant and faster than bio-processing, and might have played a more important role. The soil C is still the dominant input for the pools despite the higher level of microbial activities in water. While the authors did a good job highlighting the difference in DOC concentration and composition, but as the paper is about the discontinuity between pools and peat, it's important to better explore how the water being transported between peat and pools (even vertically), what happens at the interface, what kind of DOC is exported and why.

**Technical corrections**

21: Please change "If" to "While".

39: Please delete "net".

49: What processes of organic carbon do you refer to?

77: This paper presents a study about DOC lability from boreal peatlands with porewater sampling (https://doi.org/10.1139/cjss-2019-0154), so the authors may want to change the argument that no insight about changes in DOM composition in boreal peatlands.

141: It's not clear what monitoring "among others" refers to.

178: Both UV and fluorescence are optical analyses.

3.2.2: Is it better to shorten this part and highlight the key information, as it's effectively repeating what's in each of the graphs in Fig. SI.3.

185: What calibration was conducted after observing the difference in $Abs_{254}$?

190: It's not clear which samples were analysed with Duetta in 2019. Just 2019 samples or both years?

185-230: the description of the method details can be simplified, and information of each index presented more systematically. It's a bit lengthy with much detailed information.

233: Can just use DOM as being introduced already. Please check throughout the manuscript.

238: I'm not fully convinced this mixing was necessary. The variability can be considered in the statistical analysis. And why did the authors only mix the porewater but not the pool samples?

214: Was there additional cover for the amber glasses to completely block out the light? Did you test the light penetration through the vials?

246: Why was the porewater samples placed at the outlet instead of inside of the wells? Was it because the authors wanted to monitor the hourly temperature? In addition, the authors didn't provide information on if there was headspace in the glasses/vials, if they were open during the incubation for gas exchange.

253: Do you mean both DOC and TN were measured, or a ratio of DOC/TN was examined directly?

Table 1 and Figure 2: Is it necessary to have both in the results, as they present mostly the same results.

Fig.2: Why were there seasons with <5 samples? In the methods, it says 6 pools in 2018, 11 in 2019, and 6 wells in 2019.

Fig.3: The negative relationship mainly existed in the porewater samples, while the correlation for the combined samples was not that good with cor = -0.53. Maybe it would make more sense to look at the relationship separately, which would help highlight the different C dynamics between the two C sources.

334: There are several cases throughout the paper saying e.g. "As for SUVA$_{254}$", or 'As for the FI". Do you mean compared to the changes observed in SUVA$_{254}$? Can you refine this please?

375: DOC:Cl does not seem to be mentioned in the method. I understand the authors may have more data than presented in the paper, but please check and avoid mistakes like this.

380: Can you include the PCA analysis for the seasonal effect, maybe in supplementary information?

Fig 4: Caption was repeating the text in the results so could be shortened. Does DOC:Cl actually refer to DOC: DON? Information on R package for ellipses is not needed here but can be in methods.

387: Could delete "Statistical tests also revealed that" and replace with "In addition, the..".

391: Was the absence of filtered samples in August considered, as this could skew the difference between the two treatments?

412: The sentence needs some changes.

493: What do you mean by "low apparent degradation rate"?

505: Is this 136350m3 the volume of the pools in this study? While it's small DOM degradation in these sites, what would it be if scaling up for the whole Bouleau peatland? It may not only be 'slight effect' if considered collectively. In addition, seasonal variation in DOC concentration and degradability could also mean that in some months, the pools may act as 'hotspots' for GHG emission, which would be important information for peatland management along with global warming.

515: I may suggest an alternative next step to explore the effects from pools and peat morphology on DOC transport from peat to water, as it is not so clearly assessed yet but could be important as regulating the water and DOC sources.

---

## Author Comment (AC1)

**Response letter to the reviewers of the manuscript bg-2022-71**

In this response letter, the reviewer's comments are in ***italic bold black,*** our responses are in blue and significant new text added to the manuscript are in *italic green.* Changes made in the manuscript are tracked and referred to the revised manuscript.

**Reviewer #1 – Audrey Campeau**

***This paper presents a large and detailed investigation of DOC concentration and composition in peat 4 porewater and pools in a pristine peatland in northern Quebec. The dataset is interesting. As clearly stated in the title, the study reveals major "discontinuities" in DOC concentration and composition between porewater and pools within a peatland. This could have been done with a single spatially distributed sampling, but the authors complement this dataset by repeating the sampling over 2 different seasons. The seasonal sampling corroborates the initial findings of "discontinuity" in the DOC concentration and composition between these two environments. Whatever hydroclimatic conditions, the DOC in the pools and surrounding peat porewater seems to be considerably different. Overall, I find the dataset presented here to be interesting and the methods and statistics are sound. However, I have some concerns over the interpretation of the findings and how they support the conclusions of the study. In addition, I have made some recommendations to improve the data visualizations and some elements of the text, mainly the discussion.***

We thank the overall positive evaluation made on the submitted manuscript and the constructive received. We are pleased to note that the reviewer recognized the contribution of the multiple sampling periods. It is an important aspect of our work that we are very excited to present. The repetition of sampling periods over two growing season is, in our opinion, very important to catch both discontinuities between the peat porewater and the pools but also the temporal dynamics of both DOM concentrations and composition. In the following letter, we hope we addressed all the comments made by the reviewer on the previously submitted version of the manuscript. We hope we clarified the interpretation of our data and the conclusions.

**1) Causes of discontinuity**

***The peat reaches 4 m deep in some locations (line 106) (often near the pools based on the map in Primeau and Garneau 2021), but the porewater sampling considered only the top 2m (Line 125). I expect the reason for that is the assumption that hydraulic conductivity decreases exponentially with depth (stated in discussion 460-466). Therefore, the porewater in the bottom 2m of the peat profile is considered to move very slowly and contribute little to runoff generation or the water contained in pools. However, deep preferential flow areas exist in many***

*peatlands (often below 2m deep) (e.g. two Swedish studies DOI: 10.1111/gcb.13815, DOI: 10.1002/hyp.10300 (one where I was involved, sorry for citing myself), UK peatlands (e.g. DOI:10.1016/S0341-8162(01)00189-8) and GLAP peatlands (DOI:10.1002/2016GB005397, DOI:*

*10.1002/hyp.9983 ). These studies have shown that deep peat horizons can contribute to a large fraction of the runoff generation, or at least be hydrologically active. The high hydrostatic pressure in deep peat preferential flow areas can make that water emerge rapidly to the surface in specific locations, for example in streams or pools. Could this also be the explanation here? The water and C found in pools could in fact be feed predominantly from the bottom instead of*

*laterally? This could also explain why the water table is more stable in the pools than the peat porewater (Line 300, Fig SI2). Hence, the discontinuity in DOC concentration and composition observed here could in fact be due to that the sampling didn't capture the actual source of water and C (if it is located below 2m deep). The difference in specific conductivity between the pools and peat porewater, which here could act as an independent water tracer, indicates different*

*water sources (Line 304-306).*

*Assuming the authors finds this to be a plausible explanation, I want to highlight that I still find the results of the study to be interesting and relevant by simply highlighting the major and persistent disconnect between surface porewater and pool water. This disconnect possibly arises as a result of the complex hydrology of peatlands. Maybe the paper would benefit from*

*emphasizing this disconnect rather than to suggest a "common source" and find a reason for the apparent discontinuity (which are suggested to be a combined result of hydrological, chemical and biological process) (see my comment on the discussion). Maybe it's my background showing here, but I believe a missing water source could explain nearly all the observed patterns in this dataset.*

We thank you for this comment, suggestion and references. We examined the hypothesis of a deep-water source in pools, explaining the discontinuity we observed. We agree that our experimental design did not allow us to test the hypothesis of a deep source of water supplying DOM to pools. However, we think it is unlikely that the DOM present in pools derives predominantly from deep peat horizons – although it could still partially contribute to fuel the pools with DOM. Upward water movement in peat seems very unlikely at our site. Glaser et al. (2016) partly linked deep horizons water movements with groundwater dynamics at the watershed scale. As the site is surrounded by the Canadian Shield, the aquifers present low conductivity and is unlikely s hydrologically connected to the peat.

The work presented by Holden and Burt (2002) focused on the hydrological dynamics associated with peatland pipe. However, this feature is not present in our site. Thus, the site conditions are very different between our studied peatland and the one from Holden and Burt (2002) which makes the comparison difficult.

We included the hypothesis of upward water movements which supplying pools in DOM in the discussion and included some discussion on DOM composition in deep horizons, based on the work of Tfaily et al. 2018 (l. 471-479).

*"The surface flow path could also be supplied by a deep-water source enriched in DOM. It was shown that deep flow path (below 2 m depth) could supply the surface flow (Levy et al., 2014; Peralta-Tapia et al., 2015) This upward movement of water might transport deeper DOM to surface waters (Campeau et al., 2017) and explain the differences in DOM composition observed between peat porewater and pools as it was supplied by a deep horizon rather than lateral transport. However, the DOM composition in depth is supposed to be relatively similar to the composition in surface peat with a high aromaticity and average molecular weight (Tfaily et al., 2018). This is not comparable to the DOM composition observed in pools (Fig. 2) and suggest that this process might only partially contribute to the shift in DOM composition between environments."*

**2) The common source**

***The section 5.1 of the discussion claims that the differences in DOC concentration and composition hide a common source. The author claims that a common source to be C3 plant-derived, which leads me to wonder - what else could it have been in a boreal peatland? I have several issues with this aspect. 1. It seems obvious to me that the DOC would be predominantly plant-derived in a peatland and therefore doesn't constitute a hypothesis that needs testing nor a substantial finding out,***

One of our first goal when designing this study was to evaluate the contribution of aquatic primary production. Based on our results, the message has evolved towards stressing the importance of terrestrial markers into the pools rather than testing the hypothesis of the plant-derived DOM source into peat porewater.

We have modified the manuscript to remove the ambiguities about the DOM source and the term "hide" was removed from the title of the section 5.1 (l.409).

*"5.1 Differences in DOM concentrations and composition between peat porewaters and pools but a similar source"*

***2. The author hasn't clearly stated what other possible sources could be, I suppose these are "C4 plants", which generally are absent at this latitude or "microbial-derived" which are certainly overridden by the decomposing peat material.***

We agree that the aspect concerning DOM source needs to be clarified and have made the necessary changes throughout the manuscript. It its particularly clear that no other source than C3 plants was expected in peat porewater DOC. In pools, in situ primary productivity was considered to be a second potential source. However, primary productivity was not observed to be of substantial contribution at our site but still needed to be mentioned. We adjusted different parts when the DOM sources were mentioned:

- in the abstract (l. 21-22)

*"The molecular analyses and the DOC:DON ratio showed that DOM in pools was derived from the peatland."*

- in the introduction (l. 58-60)

*"The DOM of pools may derive from surrounding terrestrial peat (i.e., allochthonous) or be the result of their internal primary production through phytoplankton and microbial production (i.e., autochthonous)."*

- in the discussion (l. 430-431).

*"Our results indicate a dominant plant origin of DOM and reflect the dominant contribution in allochthonous DOM in pools."*

**3. The vocabulary often changes, sometimes this source is referred to as "terrestrial contribution" "vegetation origin" "plant-derived". I suggest that the other clarifies both the hypothesis tested here and the vocabulary.**

We agree that rather than use the expression of "common source", we need to emphasize that the results showed that the DOM in pools was derived from the transfer of plant-derived DOM, produced in peat. According to this statement, the terms "referring to the DOM source in pools" were changed and homogenized to "plant derived" or "plant origin" (l. 193; 430; 432; 437; 555).

**This leads me to further concerns over the interpretation of the d13C-DOC values and the correlation between d13C-DOC and DOC_DON ratio in Figure 3. The d13C-DOC values reported here varies across a narrow range (-25 to 28‰) and show that DOC originated from C3 plant metabolism. Meanwhile the C:N ratio show that the DOC is strongly terrestrial as opposed to aquatic. This is not surprising. But I doubt that anything else can be said of these values. What is the interpretation of the correlation Fig 3? the discussion mentions this correlation briefly on line 421-423 and Line 434-436 (albeit a missing reference to the figure here). The authors state that this correlation reveals that DOC comes from plant leachates instead of microbial exudates, but I don't see how this is supported. The study cited here (Magill and Aber, 2000) has no mention of the stable C isotopes. Are you suggesting that there is a fractionation process taking place here, whereby DOC becomes lighter with increasing DOC:DON ratio due to microbial or photodegradation? Below is a typical biplot from a review paper on d13C-DOC and DOC:DON across ecosystem types that puts in context your data ([https://doi.org/10.1016/j.earscirev.2005.10.003](https://doi.org/10.1016/j.earscirev.2005.10.003)).**

[Figure]

As we can observe in the shared picture, our data clearly present typical $\delta^{13}$C-DOC of both terrestrial plants and freshwater DOC as expected. However, within this range, we think it is relevant to emphasize that our sampling strategy allowed us to capture $\delta^{13}$C-DOC evolution. We observed divergent trends of $\delta^{13}$C over the growing season between the two environments. In peat porewater, an increased contribution of plant-derived DOM seems to be occurring. In pools, we think that the increasing values were related enhanced microbial processing of DOM.

Concerning the use of the DOC:DON ratio, it is worth noting that at our site, we measured DOC:DON ratio exceeding 50 in pools, outside of the range of freshwater DOC presented in the table above. The DOC:DON ratio is also important to document the variations within the pools as a slight but significant increase was observed during the growing season (Fig.2 and Table SI. 2).

We agree that the use of the correlation between DOC:DON ratio and $\delta^{13}$C is not necessary to explain the discontinuity we observed between the peat porewater and the pools. As this point, we decided to remove it.

**3) The role of biodegradation and photodegradation in peatlands**

*Are local differences in DOC lability really important for a peatland given that other environmental factors limit the metabolism in peat porewater. It's again interesting to measure the lability of DOC as a tracer of DOC sources, but other possibly more important factors limit the degradation of DOC in peat soils. The author also states that the slow hydraulic conductivity*

*increases residence time and therefore the potential transformation of DOC. This is true for most surface water environment (e.g. 10.1038/ngeo2720), but other possibly more important factor (e.g. electron acceptor availability) limit microbial metabolism in peat. I am not sure this is a relevant argument here, especially since hydraulic conductivity and water residence time were not quantified here as far as I am aware. DOC will be degraded once the environmental*
*conditions allow it, and that is possibly outside of the peatland catchment boundary.*

Concerning the degradation experiments, the goal of this approach was not to trace the sources of DOM but 1) to test the sensitivity of DOM of different environments to the main degradation processes and 2) to document how those processes can impact DOM composition. This was adjusted in the method section (l. 231-233)

*"The objective of DOM incubation experiments was to test the sensitivity of DOM to biodegradation and photodegradation and determine how it could affect its composition. The incubation experiments were designed to test the effects of temperature and total organic carbon versus dissolved organic carbon. "*

We agree that the degradation of DOM can happen downstream and outside the peatland
boundaries. However, many studies point out the degradation of DOM also occurs in peat porewater (Hutchins et al., 2017; Worrall et al., 2017). This is supported by the increase of microbial markers we observed in summer in peat porewater (see fMIC and %LMWFA in Table 1).

As the limitation of microbial metabolism that can occur in peat. The manuscript was modified in consequences (l. 481-483).

*"However, the low cations and anions concentration in ombrotrophic peatlands (Gogo et al., 2010) and the low-nutrient availability (Bengtsson et al., 2018) might limit the microbial degradation of DOM in peat porewater."*

*I find interesting that the authors quantify the potential photodegradability of the DOC to understand the possible fate of the DOC in downstream environments. But I doubt that*
*photodegradation within this peatland catchment can possibly be an important process for the overall peatland C budget, given 1. the limited amount of light penetrating in peat and pools, and 2. The small areas covered by the pools. The DOC being transported with water will eventually leave the catchment boundary and maybe then photodegradation can then play a role. I find interesting that this aspect was quantified as a way to characterize DOC properties*
*and act as another tracer of DOC sources, but the way it's presented here, in a context of a mass budget and as a possible mechanism for the disconnect in DOC between porewater and pool seems overstretched.*

The incubation experiments under sunlight exposition were a way to test the sensibility of peat porewater and pools DOM to photodegradation within the limit of the peatland catchment. This
process is known to affect DOM concentration and composition in boreal surface water (Lapierre and del Giorgio, 2014) and DOM composition in Arctic peatland thaw pools (Laurion and Mladenov, 2013). The key message of our experiment was the absence of sizeable effect of photodegradation on DOM composition and concentration. According to the absence of significant differences between the condition of biodegradation and photo + biodegradation, the average degradation rate was used in the mass budget we calculated (l. 527). As the photodegradation do not appear to be an important process in DOM concentration differences between peat porewater and pools, we did not discuss the importance of this process in the context of mass balance. It was discussed in the context of differences in DOM concentration and composition between peat porewater and pools for the peatland net C budget.

**4) The influence of DOM adsorption**

***The peat porewater samples were collected through a PVC tube covered with a nylon sock. I would assume that the porewater DOC that is adsorbed to the peat to not sampled then. So can this mechanism really be important to explain the discontinuity between pool and porewater DOC?***

We agree that the sampling method used to collect peat porewater potentially exclude a fraction of DOM which could have been adsorbed to the PVC tube. The water collected is the one mobile into peat and thus potentially transferable through the pools and we modified the text in consequence in the method section (l. 126).

*"This method allows to collect the mobile water which circulate into the peat."*

The discontinuity was also observed when the peat-pool gradient was performed in 2019 using another sampling method (detailed in the SI). The sampling method was not through PVC tubes but by applying a depression with a peristaltic pump and collected through laboratory-grade plastic tubes. The DOM sampled with this method supported the trends we observed for DOC concentrations, DOC:DON ration, SUVA$_{254}$ and E2:E3 ratio, confirming the limiting effect of adsorption processes. However, a limitation of this sampling method is that it was harder to sample porewater at depth below 100 cm into peat.

**5) Adjustments of Data Visualization**

***Figure 2 is a key figure presenting the data, but I find the box plot to be an ineffective choice of visualization in this case. There are too many plots and too many things being compared for this type of plot to work. A more effective visualization could be a parallel coordinate plot (e.g. https://datavizcatalogue.com/methods/parallel_coordinates.html). Each vertical axis would represent a different variable (e.g. DOC concentration, DOC:DON etc.), and each line moving laterally would be a different sample location. I would suggest to fade the peat porewater samples in the background and superimpose the pool water samples on top in a darker color to help compare these two environments. You could even make it a three panel figure, one for each season. This would allow to see at a glance, which site/season bear most similarities and differences for all variables. If possible, maybe also indicate the meaning of optical properties on the axis, for example (higher SUVA values is more aromatic and lower is less aromatic etc.).***

*This would facilitate the visual interpretation. This plot would also give the possibility to merge*

*figure 5 and figure 2, in this case by adding another vertical axis for degradation rates.*

*If you choose to stick to the boxplot format please add letters on the x-axis to show statistically different groups. (e.g. function multcompLetters in R, library multcompView)*

We thank the reviewer for the visualization suggestion . We took the comment into account and obtained this new version of the Fig. 2. We did a parallel coordinate plot with color represents the environments and the line-type the season for centered-reduced means of variables considered in the previous version of the figure.

[Figure]

However, the conception of the figure was challenging, and we were forced to apply a data transformation (centered-reduced means) and to represent average values per environments and per seasons. For this reason, we consider we loose too much information and we decided to keep the first version of the Fig. 2. In the Fig. 2, letters were added to identify the statistical differences between groups (seasons and environments).

[Figure]

*Can the symbols in the PCA (Figure 4) be the same at in Figure 6? Also note that all symbols in figure 6 are circles so there is an error here. The figure 4 could also be bigger for better readability.*

The size of the fig. 4 was changed and the symbols were homogenized with the fig. 6. In the Fig. 6, all shapes were changed for circles as the season was not a factor we discussed.

[Figure]

[Figure]

**6) Line by line comments**

*Line 69 to 78: Those lines might fit better in the discussion if you choose to emphasize the differences in DOC sources between porewater and pools.*

We think that this section is important in the introduction as it presents how the differences that have been observed between peat porewater and pools can be the consequences of many processes we actually refer to later in the discussion.

*Line 383: Is it the "average" or "median" degradation rate that was statistically significantly different?*

As the statistical test was ANOVA, it is the average that was is compared. The text was modified (l. 381).

*"Statistical tests revealed no significant differences in the average degradation rates between in*
*situ and controlled conditions of biodegradation (section 3.5.1)."*

*Line 410: Be more specific here. Is it the average or range in DOC, porewater or pool water? It can also be helpful for the reader that you write in bracket the number you are referring to, even if they are available in the table SI3.*

The text was modified accordingly, and the term *average* was added (l. 410).

*"The average DOC concentrations measured in peat porewater at our sites during the growing season are in agreement with the expected range of a subarctic peatland (13.9-28.8 mg L-1; Deshpande et al., 2016)."*

*Line 412-414: Do you mean here that the subarctic and boreal peatlands have on average 20 mgCL less DOC than temperate ones? Be more clear about what you are comparing here. Also, sentences starting with " A synthesis … " or " a study …" make the text more tedious. You can go straight to the point here and say for example. Porewater DOC concentration in peatlands exhibit a strong latitudinal trend, whereby boreal and subarctic peatlands contain …". Also, its not clear why this study is mentioned here and what argument you are trying to make. Is this just to state that your DOC concentration are "normal" or are you trying to extrapolate your findings to other latitudes*

The text was adjusted and simplified in the first paragraph of the section 5.1, thanks to the comment (l. 410-415).

*"The average DOC concentration measured in peat porewater at our sites during the growing season is in agreement with the expected range of a subarctic peatland (13.9-28.8 mg L-1; Deshpande et al., 2016). The DOC concentrations in peatland peat porewaters exhibit a latitudinal gradient, from DOC concentrations commonly lower than 20 mg L-1 in boreal and subarctic latitudes compared to temperate latitudes during growing seasons (Table SI.4). This observation, in line with our results, suggests a temperature control on the balance between DOM production and processing (Kane et al., 2014)."*

*Line 421: Instead of comparing with the name of the study, you could refer to the type of peatland that was studied in this paper Also, why are you making this comparison, again the argument is missing here: The DOC:DON ratios measured in peat porewaters at our study site were up to six times higher than in Austnes et al. (2010) " suggesting that …".*

The text was modified and now refers to the study site's region rather than the paper cited. Also, the text was modified, and an argument supporting the use of this reference was added to justify the reference cited (l. 421).

*"The DOC:DON ratios measured in peat porewaters at our study site were up to six times higher than in a Welsh temperate peatland (Austnes et al., 2010). These high ratios suggest a strong signature of plant leachates in peat porewater DOM composition."*

*Line 452: This is a hypothesis here, no? The photodegradable fraction of DOC might have already been degraded prior to sampling, but the way it's written here makes it sound like you are certain that's the case.*

We agree that the phrasing might be confusing. The message here is that we hypothesize that the higher aromaticity we observed during the punctual sampling is in line with the increase of $SUVA_{254}$ observed during the incubations and support the biodegradation of DOM in peat porewater. The text was modified in consequence (l. 451-456).

*"The absence of sizeable photodegradation suggests that this process did not drive DOM composition in pools. The clear pattern of the SUVA254 increase observed during the incubation period was independent of the conditions where DOM was exposed to solar radiation. This is coherent with the biodegradation of non-aromatic molecules (Spencer et al., 2008, 2015; Mann et al., 2015; Worrall et al., 2017) leading to an increase of SUVA254 (Hulatt et al., 2014; Autio et al.,*

*2016) while photooxidation has been shown to induce a decrease of DOM aromaticity (Laurion and Mladenov, 2013; Ward and Cory, 2016)."*

**Line 453: By consumption here you are referring to the biological pathway, not the photochemical one. Please clearify.**

Indeed, it is biodegradation. The text was corrected (l. 453).

*"This is coherent with the biodegradation of non-aromatic molecules (Spencer et al., 2008, 2015; Mann et al., 2015; Worrall et al., 2017) leading to an increase of SUVA254 (Hulatt et al., 2014; Autio et al., 2016) while photooxidation has been shown to induce a decrease of DOM aromaticity (Laurion and Mladenov, 2013; Ward and Cory, 2016)."*

**Line 462: can they "be explained by" or they can "arise as a result of". This sounds like you are pleading a case for more "positive" result, while a more "negative" result in this case can be even more interesting.**

That is an excellent remark. We tried to find a better formulation and the sentence was changed as follows (l. 459-460).

*"The observed differences in DOM composition between peat porewaters and pools and its persistence during the growing season was driven by a combination of hydrological, chemical, and biological factors."*

**Line 463: That doesn't mean that water cannot be constantly filled from the bottom and just occasionally sourced from surface peat when the water table is high.**

We modified the text according to the general comment you made about the upward contribution of water and DOM (l. 469-477).

*"The surface flow path could also be supplied by a deep-water source enriched in DOM. It has been shown that deep flow path (below 2 m depth) could supply the surface flow (Levy et al., 2014; Peralta-Tapia et al., 2015) This upward movement of water might transport deeper DOM to the surface waters (Campeau et al., 2017). This movement of water might explain the differences in DOM composition observed between peat porewater and pools as it was supplied by a deep horizon rather than lateral transport. However, the DOM composition in deep layers is supposed to be relatively similar to the composition in surface peat with a high aromaticity and average molecular weight (Tfaily et al., 2018). This is not comparable to the DOM composition observed in pools (Fig. 2) and suggest that this process might only partially contribute to the shift in DOM composition between environments."*

**Line 410: The reference to table SI3 should be placed at the end of the sentence.**

The text was changed (l. 413).

*"The DOC concentrations in peatland peat porewaters exhibit a latitudinal gradient, from DOC concentrations commonly lower than 20 mg L-1 in boreal and sub arctic zones compared to temperate zones during growing seasons (Table SI.4)."*

***Line 410-415: Why mention this latitudinal effect? There seems to be an argument missing here. Do 191 you mean that your data fit in with other peatlands at the same latitude? Please clarify.***

This part was for contextualized the results at our study site. We hope that the modifications made in this section make this paragraph clearer (l. 410-415).

*"The average DOC concentrations measured in peat porewater at our sites during the growing season are in agreement with the expected range of a subarctic peatland (13.9-28.8 mg L-1; Deshpande et al., 2016). The DOC concentrations in peatland peat porewaters exhibit a latitudinal*

*gradient, from DOC concentrations commonly lower than 20 mg L-1 in boreal and sub arctic zones compared to temperate zones during growing seasons (Table SI.4). This observation, in line with our results, suggesting a temperature control on the balance between DOM production and processing (Kane et al., 2014)."*

***Line 507-517: If the pools allow old DOC to make its way to the surface and therefore enter the***

***peatland contemporary C cycle then they become very important for the stability of the old peat C stock. But based on the molecular weight indexes, the DOC seems younger in the pools than porewater***

We modified the text according to your comment (l. 532-536).

*"Also, the DOM in pools presents characteristics of recently produced DOM transferred from peat*

*so its biodegradation might not affect the carbon from old peat horizons as the DOM biodegradation seems to be particularly influenced by temperature (Fig. 4) and DOM production (Fig. 2) during the growing season. The biolability of DOM has to be put in the perspective with the peatland carbon budget as it could be impacted by interannual climatic variations and their future trends. "*

***Line 520: My personal suggestion for future studies would have been to combine studies on DOC cycling with interdependent water tracer. It's hard to trace back the cycling of DOC without knowing the water source.***

According to your different remarks regarding the importance of the hydrology in the discontinuity, a sentence was added to the conclusion (l. 561-563).

*"As the dynamics of DOM in peat porewater seems closely connected to the hydrology of the peatland, it seems important to better identify it with the water source and its circulation through the peat."*

***Line 522: Exactly! I would even add, that the disconnect between the two environments persist no matter what the hydroclimatic conditions are.***

We rephrase this part to emphasize the point you mentioned (l. 557-558).

*"This study highlighted that DOM is a highly dynamic component of the carbon cycle in peatlands, with important changes identified in its concentration and composition in both peat porewaters and pools. This discontinuity was persistent throughout the growing season and different hydroclimatic conditions."*

*Line 525: Or is the concentration just increasing because it gets drier? Maybe check if this difference persists once the DOC is volume-weighted based on water table position?*

The hypothesis of increase in DOM production is based on the postulate that higher temperatures during summer are known to influence DOM production in peat porewater (Laudon et al., 2012; Grand-Clement et al., 2014; Zhu et al., 2022).

*Line 534: the term "physicochemical parameters" is vague.*

The text was refined (l. 572).

*"The rapid modification of physicochemical conditions (e.g., temperature and oxygen availability) between those environments might favour the biodegradation of DOM at the interface between the peat and the pools and within the pool."*

*Line 535: What you say here is true, but I find it to be a disappointing end to a paper. The last sentence of the conclusion is very important, and I am sure there is more a interesting take home message to be given here.*

We proposed a better take home message centered around the discontinuity (l.571-574).

*"The strong discontinuity of DOM concentration and composition observed between the peat*
*porewater and pools and its persistence during the growing seasons, and under different hydroclimatic conditions emphasize the significance of small spatial scales processes."*

*Table SI.3: DOC mean and SD?*

This is mean and SD and the text was corrected.

*Fig S1: If E2:E3 and Sr are both proxies of molecular weight (Line 189), why is it that they*
*correlate so poorly?*

This is mainly because even they are both proxies for molecular weight, they are associated with different fractions of DOM. The E2:E3 is also negatively correlated with the DOM aromaticity (see the negative correlation with the $SUVA_{254}$ in Fig. SI.1) while the Sr is also associated with the microbial metabolism (see the positive correlation with the deoxyC6:C5
and the %CARMIC in the Fig. SI.1).

---

## Author Comment (AC2)

**Response letter to the reviewers of the manuscript bg-2022-71**

In this response letter, the reviewer's comments are in ***italic bold black**,* our responses are in blue and significant new text added to the manuscript are in *italic green*. Changes made in the

5    manuscript are tracked and referred to the revised manuscript.

**Reviewer #2 – Anonymous**

We would like to thank the reviewer for the evaluation of our work and for the constructive comments . We appreciate that the reviewer recognized the importance to document the DOM dynamics in peatlands. We have tried to address the remarks and clarify some aspects of the

10   manuscript. The line-by-line comments have been integrated to the text.

**1)   Regarding the Mat & Meth section**

***In general, I think the writing of method is a bit lengthy. While it's good to provide such information for readers who want to replicate the method, there are too many details which are not necessary to be included in the main text of the paper. I'd like to suggest the authors to***

15   ***refine this part in a concise manner, combining with references and supplementary materials.***

***In addition, as various sampling trips, analyse methods, proxy indices are used in this study, I'd like to suggest using a table or diagram to summarise this information, which would make it much easier for the readers to understand the research design and interpretation. For example, how many samples from what sites on which dates, and which were analyzed for what. It'd also***

20   ***be helpful for the readers to understand why dot plots were used in Fig. 2.***

We tried to consider reducing the material and methods section. However, numerous methods were used, explaining the density of this section. Also, the detailed method could be beneficial for further studies.

To help the understanding of the sampling and analyses design, a synthesis table with number of

25   samples and analyses performed were realized. It is added to the supplementary information (Table SI.1) and was referred to in the Material and Method section (l. 132).

*"Samples collected per campaigns and analyses performed on it were synthesized in table SI.1."*

Table SI.1. Synthesis of samples number per analyses per campaigns.

| Year | Campaign | Environment | Analyses | | | | | |
|---|---|---|---|---|---|---|---|---|
| | | | DOC; DOC:DON | Isotopic | Absorbance | Fluorescence | Molecular | Incubation |
| 2018 | June | Porewater | 5 | 1 | 5 | | | |
| | | Pools | 6 | 2 | 4 | | | |
| | July | Porewater | 4 | 2 | 4 | | 3 | |
| | | Pools | 6 | 3 | 6 | | 3 | |
| | August | Porewater | 6 | | 6 | | 3 | |
| | | Pools | 6 | | 6 | | 3 | |
| | September | Porewater | 4 | 2 | 4 | | 3 | |
| | | Pools | 6 | 3 | 6 | | 3 | |
| | October | Porewater | | | | | | |
| | | Pools | 6 | 2 | 6 | | | |
| 2019 | June | Porewater | 6 | 3 | 6 | 6 | 2 | x |
| | | Pools | 11 | 3 | 11 | 11 | 2 | x |
| | August | Porewater | 6 | 3 | 5 | 5 | 1 | x |
| | | Pools | 11 | 3 | 11 | 11 | 3 | x |
| | September | Porewater | 5 | 5 | 5 | 4 | 3 | x |
| | | Pools | 11 | 3 | 11 | | 2 | x |
| | October | Porewater | 5 | 3 | 5 | 5 | 3 | |
| | | Pools | 5 | 3 | 5 | 5 | 3 | |

30

**2) Differences in DOM Concentration and Composition**

*Firstly, when comparing the DOC concentrations in porewater and pools in this study with those in other climatic zones, seasonality should be considered, as here DOC samples were collected in the growing seasons which would tend to be higher than other seasons.*

35 Thank you for this remark. It is true that we did not mention that our study presented data sampled during the growing season, which may not be the case for all compared studies. After checking, we observed that sampling periods of all studies except two occurred during the growing seasons (Beer and Blodeau, 2007; Tipping et al., 2010). We annotated those two references and adjust the text in consequence (l. 410; 413; legend of the table SI.3).

40 *"The average DOC concentrations measured in peat porewater at our sites during the growing season are in agreement with the expected range of a subarctic peatland (13.9-28.8 mg L-1; Deshpande et al., 2016)."*

*"The DOC concentrations in peatland peat porewaters exhibit a latitudinal gradient, from DOC concentrations commonly lower than 20 mg L-1 in boreal and sub arctic zones compared to*
45 *temperate zones during growing seasons (Table SI.4)."*

*Secondly, the authors only explained the good correlation between DOC:DON and 13C in porewater, but didn't try so with the pool DOC. The absence of this correlation in the pools could well lead to the discussion in 5.2, and highlights the discontinuity of DOC composition.*

Your comments about the use of the correlation between DOC:DON ratio and the d13C DOC ratio points out that it is not necessary to document the discontinuity between peat porewater and pools. We think that the use of the DOC:DON ratio and the d13C DOC ratio independently were more pertinent to discuss the terrestrial source of DOM in pools (according to the high DOC:DON ratio measured in pools) and discontinuity of peat porewater and pools DOM (according to the divergent seasonal trends in d13C DOC). As suggested, and in line with the other reviewer's comments, this figure has been removed.

*Lastly, while the photo-degradable DOC might have been quickly degraded in the first few days, I'm not entirely convinced that it would be the main reason. DOC from porewater was not exposed to light before being collected so there should be minimal effects from photodegradation. In pools, new DOC inputs would be expected with the increased precipitation, which was observed especially during the summer and autumn seasons. Therefore, there could be continuous supply of photo-degradable DOC during those periods of time. Furthermore, in boreal and arctic areas, the amount of sunlight is less abundant than low latitude areas, limiting the photodegradation of DOC, although I realised it would be less so in summer. Did you have any data on the light? Was any incubation conducted when it was rainy or cloudy? Did the glasses/vials filter out certain wavelengths which cause photodegradation? As it stands now I don't think there is enough evidence to make the argument that there was no photodegradation process in the samples.*

Considering photodegradation, we were also surprised when we found out that the DOM photodegradation was not sizeable despite being considered to be a common process in other boreal surface inland waters (Lapierre and del Giorgio, 2014; Jones et al., 2016).

Concerning the amount of sunlight, the study site it at 51° of latitudes and in summer, sunlight duration is longer than 12 h day$^{-1}$ and the photosynthetically active radiation is higher than close to the equator during summer months (see the figure A bellow from Mortensen, 2014) and the downward shortwave radiation increase during summer months in the north hemisphere (see the figure B bellow from Hatzianastassiou et al., 2005). Thus, we believe that sunlight is not limiting in the period of incubation at our study site.

A) Mortessen, 2014

[Figure]

Figure 1. Photosynthetic active radiation (PAR) at different latitudes inside a greenhouse (60% transmission) throughout one year, given as daily means per month [1] [2]. A conversion factor of 7.9 mol per 1.0 kWh was used.

B) Hatzianastassiou et al., 2005 (the star represents the approximative location of the study site)

[Figure]

Fig. 2. Long-term (1984–1997) average global distribution of net downward (or absorbed) shortwave radiation (in $Wm^{-2}$) at the Earth's surface for the mid-seasonal months of (a) January, (b) April, (c) July, and (d) October.

To address the comment related to the potential effect of clouds and rain on light exposure variability between the incubation periods, we have looked at the incoming radiation based on a weather station installed at our study site (photosynthetically active radiation using a LI-190R and shortwave incoming radiation using a CNR4). During incubations in August, average photosynthetically active radiation (PAR) was 37.9 mol m$^{-2}$ day$^{-1}$ and the average shortwave incoming radiation (ISWR) was 346.8 W m$^{-2}$ for an average sunlight duration of 15.5 h/day. In September, the PAR was 34.3 mol m$^{-2}$ day$^{-1}$ and the ISWR was 329.5 W m$^{-2}$ for an average sunlight duration of 13.6 h/day. Unfortunately, data was not available for the incubation in June, but during the 6 days before the incubation, the PAR was 48 mol $^{-2}$ m$^{-2}$ day$^{-1}$ and the ISWR was 422.8 W m$^{-2}$ for an average sunlight duration of 16.6 h/day. This suggests that the light conditions during our experiment where similar between one period to another and that cloudy or rainy days do not seem to have affected the overall incoming radiation.

The incubation performed in controlled conditions of temperature and darkness was a way to control the efficiency of amber glass vials to filtrate the light. No significant difference in degradation rates were observed between incubations of amber glass vials in controlled and in situ and with incubation of clear glass vials in situ conditions either as you can see in the figure below.

[Figure]

In addition, incubation performed in a solar incubator at a sunlight exposition equivalent to 6 days of natural sunlight in summer did not reveal significant changes in DOC concentrations, supporting the observation we made with *in situ* photodegradation incubations.

[Figure]

We changed "the DOM was potentially not photolabile" to "the experimental design did not allow the observation of any photodegradation"(l. 445 and the text was adjusted l.451).

*"Our data did not evidence any photodegradation during DOM incubation in peat porewaters and pools, suggesting that the DOM photodegradation was not sizeable by our experimental design."*

*"The absences of sizeable photodegradation suggest that this process did not drive the DOM composition in pools."*

**3) Processing leading to the difference**

*Firstly, the difference in DOC composition, e.g. SUVA254, E2:E3, could be a result of that DOC in peat soil is 'older' than those exported to water, which was not considered in the paper before thinking about the more instant changes caused by hydrological, chemical and biological processes as the authors focused on.*

According to the literature, it seems unlikely that the differences in composition between the peat porewater and the pools were due to different ages between the two environments. Several studies have shown that DOM is derived from fresh organic matter produced at the surface (Tipping et al., 2010; Tfaily et al., 2018). The DOM is mostly recent in peatland (Campeau et al., 2018) and older DOM was only observed in deeper horizons and occasionally mobilized through hydrological processes (e.g., mobilization during extreme storm events) and in degraded peatlands (Dean et al., 2019).

*Also, the aromatic DOC should be more hydrophobic rather than hydrophilic. If aromatic compounds are hydrophilic, it should be easier for them to flow from peat to pools leading to higher DOC aromaticity in the water, which is contrasting to what was observed.*

In our study, we observed significant higher aromaticity of DOM in peat porewater compared to pools. We hypothesized that aromatic DOM is more hydrophobic, which could enhance the exchanges between DOM and partially degraded peat (mentioned in the preprint bg-2022-71 at the l. 472-475). This point supports the idea that DOM was partially retained into peat during its

125    transfer from peat to pools, and partly explains why the aromaticity of DOM decreased from peat porewater to pools.

*In addition, in the abstract the authors pointed out the transformation of DOC at the interface led to the production of low molecular weight compounds, which is contrasting to their suggestion that microbial processing would cause the increase in aromatic DOC which is often*

130    *larger in molecular weight.*

The reviewer noticed that we mentioned in the abstract that the DOM microbial process at the interface between peat porewater and pools. This was leading to the production of lower DOM molecular weight compounds AND that we observed an increase in DOM aromaticity during DOM incubation experiments, which is expected to have a larger molecular weight. It is important to

135    understand the DOM in a sum a complex molecular mix. For example, we observed during the growing season an increase in the DOM aromaticity and a lowering of the DOM molecular weight in pools and this was observed simultaneously (Fig. 2 based on SUVA254 for the aromaticity and E2:E3 ratio for the DOM molecular weight). A diversity of DOM compounds were biolabiles and might lead to the production of different degradation products with different properties. As the

140    DOM mix is dominated by lignins, the increase of DOM aromaticity during incubation might be explained by the experimental design that could stimulate their degradation. Also, as we hypothesized that DOM aromatic compounds might be partially retained into peat, it is not contradictory that DOM compounds that were actually transferred into pools were less aromatic, and their degradation productions are less aromatics as well.

145    In line with this explanation, degradation of DOM compounds occurs at different time scales from few hours to years after its solubilization. For example, some compounds like carbohydrates are preferentially degraded while degradation of aromatic compounds might be longer. This may be why, after the six days of incubation experiments, we observed an increase in the aromaticity. These observations could have been different if measurements were made after the first day of

150    incubation. Hence, the lower DOM molecular weight could result of the rapid degradation of compounds for which the degradation products present a lower molecular weight.

*Secondly, DOC does interact with different materials or minerals in peat. For example, the oxidation/reduction of Fe have been observed to be mediated by microbes and affect the solubility of DOC (Mladenov et al., 2010), and tend to coagulate with high molar mass DOC*

155    *(Ritson et al., 2014). At the interface between peat and pools, particularly when water table is higher in peat, the change from anaerobic to aerobic environment could affect the reduction/oxidation of certain relevant minerals (e.g. Fe) and reduce mobility of certain group of DOC (Nierop et al., 2002).*

We have investigated the potential interactions between the DOM and solid and dissolved mineral in peat and peat porewater have been considered. However, boreal ombrotrophic peatlands are relatively poor in mineral elements (%C LOI = 46.7 ± 6.4, Primeau and Garneau, 2021). Then interaction with minerals in peat are limited. Regarding interaction within peat porewater, we collected some data on total trace metals concentration in 2018. Dissolved Fe was about 10 µg $L^{-1}$ . these concentrations are very low compared to the values in the suggested publication (Mladenov et al., 2010), where Fe concentrations ranged between 5 and 10 mg $L^{-1}$ but in a tropical region. The DOC:Fe even decreases from 1080:1 in peat to 62:1 in pools highlighting opposing trends between peat and pools where the Fe seems relatively soluble. The potential of DOM coagulation resulting of interaction with dissolved Fe seems negligible according to the low concentrations in dissolved Fe we measured in both peat porewater and pool.

*Lastly, it was great to see that the authors were trying to explore the biogeochemical processes from porewater, interface and pools, which could be a highlight of this paper. However I'm not entirely sure how big role microbial processing is at the interface as the authors claimed. Indeed, biodegradation could happen within a couple of days, but at the interface I tend to think the physical and chemical interactions, e.g. precipitation and binding via the changes in the physical environment from soil to water is more instant and faster than bio-processing, and might have played a more important role. The soil C is still the dominant input for the pools despite the higher level of microbial activities in water. While the authors did a good job highlighting the difference in DOC concentration and composition, but as the paper is about the discontinuity between pools and peat, it's important to better explore how the water being transported between peat and pools (even vertically), what happens at the interface, what kind of DOC is exported and why.*

As mentioned by the reviewer, the goal of the paper is to explore the various ways and processes which can lead to the discontinuity in DOM composition between peat porewater and to pools, within the framework of the sampling and analyses. The paper illustrates well the complexity of the DOM composition and the processes involved. Through this complexity, we tried to identify which processes can explain the discontinuity between peat porewater and pools. Our work led to the hypothesis that biological processes, through microbial degradation, is one of the explanations of the differences we observed. However, we would emphasize that as it is a sum of processes, we cannot totally exclude other processes that can play an important role and are mentioned in the paper. The manuscript was reworked thanks to the comments made by the reviewer. We emphasized and clarified some aspects of our discussion, mostly through the section 5.2 for this particular comment (l. 469-477).

*"The surface flow path could also be supplied by a deep-water source enriched in DOM. It has been shown that deep flow path (below 2 m depth) could supply the surface flow (Levy et al., 2014; Peralta-Tapia et al., 2015) This upward movement of water might transport deeper DOM to the surface waters (Campeau et al., 2017). This movement of water might explain the differences in DOM composition observed between peat porewater and pools as it was supplied by a deep horizon rather than lateral transport. However, the DOM composition in deep layers is supposed to be relatively similar to the composition in surface peat with a high aromaticity and average*

200 *molecular weight (Tfaily et al., 2018). This is not comparable to the DOM composition observed in pools (Fig. 2) and suggest that this process might only partially contribute to the shift in DOM composition between environments."*

**4) Technical corrections**

***21: Please change "If" to "While".***

205 The text was modified (l.20).

***39: Please delete "net".***

The text was modified (l.39).

***49: What processes of organic carbon do you refer to?***

The text was adjusted, and processes are now mentioned (l. 50).

210 *"While most studies of peatland carbon dynamics have focussed on terrestrial microforms (Nungesser, 2003; Pelletier et al., 2011; Shi et al., 2015; Chaudhary et al., 2018; Graham et al., 2020), the composition and processes of production and degradation of organic carbon in pools remain poorly documented."*

***77: This paper presents a study about DOC lability from boreal peatlands with porewater***
215 ***sampling (https://doi.org/10.1139/cjss-2019-0154), so the authors may want to change the argument that no insight about changes in DOM composition in boreal peatlands.***

We are thankful for the reference suggestion. But as the reference presents a study that takes place in a site affected by permafrost, the text was adjusted in consequence to be more consistent with our study site which was not affected by permafrost (l.78).

220 *"Studies investigating the changes in DOM composition in peatland porewaters and pools have mostly been focused on temperate (Banaś, 2013; Arsenault et al., 2019), subarctic, and Arctic regions (Laurion and Mladenov, 2013; Deshpande et al., 2016; Burd et al., 2020; Payandi-Rolland et al., 2020; Laurion et al., 2021; Moody and Worrall, 2021), but there is no insight about changes in DOM compositions in boreal peatlands non affected by permafrost."*

225 ***141: It's not clear what monitoring "among others" refers to.***

This formulation was actually not clear and removed from the text as it does not give any pertinent information (l. 144).

***178: Both UV and fluorescence are optical analyses.***

The title of the section 3.4.2 was corrected in consequence.

230 ***3.2.2: Is it better to shorten this part and highlight the key information, as it's effectively repeating what's in each of the graphs in Fig. SI.3.***

The section was shortened and only the pertinent part of the text was kept (l. 145-149).

*"Samples from the two studied years were pooled according to seasons. In this study, seasons were defined based on air and water temperatures measured at the site (Fig. SI.3). Spring was defined from the end of the seasonal thaw that occurred in May to the end of June. Summer included the months of July and August when air and water temperatures were at their warmest. Finally, the autumn season corresponded to the months of September and October when air and water temperature decreased to zero."*

**185: What calibration was conducted after observing the difference in Abs254?**

As the differences between the two methods were very low and there were no significant differences between years for absorbance index, we decided to keep it without any post-calibration. This is mentioned in the text (l. 181).

*"As no significant effect was observed between years on absorbance indices, no correction was performed on absorbance spectra."*

**185-230: the description of the method details can be simplified, and information of each index presented more systematically. It's a bit lengthy with much detailed information.**

The reviewer can refer to the comment we made on the first section 1) of the present response letter.

**233: Can just use DOM as being introduced already. Please check throughout the manuscript.**

The first mention of DOM is l.16 in the Abstract and l.51 in the introduction.

**238: I'm not fully convinced this mixing was necessary. The variability can be considered in the statistical analysis. And why did the authors only mix the porewater but not the pool samples?**

The mixing was performed because the quantity of water in piezometer was limited and not sufficient to perform all incubation conditions. The text was corrected in consequence in the methodology section (l. 238-239).

*"This strategy was used because the quantity in piezometer was limited and not sufficient to perform all incubation conditions."*

**214: Was there additional cover for the amber glasses to completely block out the light? Did you test the light penetration through the vials?**

The opacity of amber vials was not tested. However, the reviewer can refer to the comment we made in the section 2) of the present letter. We mentioned that no significant differences were found between the incubation in dark condition and in sunlight condition for amber vials.

**246: Why was the porewater samples placed at the outlet instead of inside of the wells? Was it because the authors wanted to monitor the hourly temperature? In addition, the authors didn't provide information on if there was headspace in the glasses/vials, if they were open during the incubation for gas exchange.**

The porewater samples were placed at the stream outlet for different reasons. Firstly, because the vials did not fit in wells. Secondly, to test the effect of photodegradation and to monitor the temperature. Finally, because those incubation were also performed in the stream, then the incubation of pore water in the stream simulate the transfer of peat porewater DOM to surface water.

As the bottle used were 125 mL and 100 mL were incubated, a headspace of 25 mL was kept. The bottles were closed.

Thank you for noticing these omissions, and the text was modified (l. 241-243).

*"Amber glasses of 125 mL were used to test biodegradation (BIO) only and transparent vials of 125 mL were used for bio and photodegradation (BIO+PHOTO). Each condition was incubated in triplicates with a headspace of 25 mL and bottles were tightly closed."*

**253: Do you mean both DOC and TN were measured, or a ratio of DOC/TN was examined directly?**

It was DOC and TN measured and the text was corrected (l. 352).

*"All samples (n = 36) were prepared for DOC, TN and inorganic N quantification, and absorbance analyses, before and after the incubation experiments."*

**Table 1 and Figure 2: Is it necessary to have both in the results, as they present mostly the same results.**

We think that both figure and table are complementary as Table 1 present most of the indices derived from analyses while Fig. 1 present the key results.

**Fig.2: Why were there seasons with <5 samples? In the methods, it says 6 pools in 2018, 11 in 2019, and 6 wells in 2019.**

Some analyses were not performed systematically on all samples. While it was mentioned from THM-GC-MS analyses (l. 198-199) the omission was corrected for stable isotopes analyses (l. 165-166). The table added in SI. Thanks for the recommendation that will help the understanding.

*"Analyses of δ13C-DOC were realised on 41 samples selected from peat porewater (n = 20) and pools (n = 21; Table SI.1) at the Jan Veizer stable isotope laboratory (University of Ottawa, Canada) following the method developed by Lalonde et al. (2014)."*

**Fig.3: The negative relationship mainly existed in the porewater samples, while the correlation for the combined samples was not that good with cor = -0.53. Maybe it would make more sense to look at the relationship separately, which would help highlight the different C dynamics between the two C sources.**

As it was previously mentioned, at this point we decided to remove the figure.

***334: There are several cases throughout the paper saying e.g. "As for SUVA254", or 'As for the FI". Do you mean compared to the changes observed in SUVA254? Can you refine this please?***

305     The text was refined where this kind of formulation was written (l.334, l.349 and l.367).

*"Compared to SUVA254, the E2:E3 ratio showed no significant trends in peat porewaters, but it slightly increased in pools from 4.02 ± 0.11 in spring to 4.41 ± 0.18 in autumn, suggesting a decrease of the average molecular weight during the growing season (Fig. 2.e)."*

*"As the changes observed for the FI, variations of the β:α index were limited to a small range."*

310     *"Comparatively to the variations observed for the C/V ratio, fVEG remained almost stable in pools, while it decreased in peat porewaters in autumn."*

***375: DOC:Cl does not seem to be mentioned in the method. I understand the authors may have more data than presented in the paper, but please check and avoid mistakes like this.***

The mention of DOC:Cl was removed.

315     ***380: Can you include the PCA analysis for the seasonal effect, maybe in supplementary information?***

This is the same figure as the one presented in the paper but with colour separation according to the season and no emerging trend. We are not convinced of the pertinence of this figure given the very small weight of the season in the PCA.

320     ***Fig 4: Caption was repeating the text in the results so could be shortened. Does DOC:Cl actually refer to DOC: DON? Information on R package for ellipses is not needed here but can be in methods.***

The text was shortened in the caption of the figure.

*"Figure 3. Representation of the first two dimensions of principal component analysis (PCA) of a)*
325     *physicochemical, quantitative, and qualitative parameters as variables and b) individuals."*

***387: Could delete "Statistical tests also revealed that" and replace with "In addition, the..".***

The text was simplified according to the comment of the reviewer.

*"The degradation rates were significantly higher for the incubation conditions of unfiltered samples (UF) compared to filtered sample (F) conditions (Fig. 4)."*

330     ***391: Was the absence of filtered samples in August considered, as this could skew the difference between the two treatments?***

We are very thankful that this hypothesis has been pointed out. After refining the statistical analyses, no significant differences were found between the degradation rates under filtered and unfiltered filtration a) for samples of spring and autumn season only in peat porewater and pools

335     (AOV, F = 2.631, p-value = 0.11). However, filtered and unfiltered conditions were significantly different in peat porewater only for all seasons (Welsh AOV, statistic = 6.04, p-value = 0.02).

We adapted the paper accordingly, but we kept the figure per Filtered and UnFiltered condition as the significant differences are still conform when peat porewater was grouped for all seasons (l. 396-397) and adjustment was made in the discussion (l.520-522).

340    *"After excluding the UF condition of August, there was no persistent significant differences between F and UF conditions."*

*"Degradation rates under unfiltered conditions in pools were two times lower than for peat porewaters and no significant differences were observed in spring and autumn."*

**412: The sentence needs some changes.**

345    This was noticed and the text was modified.

*"The DOC concentrations in peatland peat porewaters exhibit a latitudinal gradient, from DOC concentrations commonly lower than 20 mg L-1 in boreal and sub arctic zones compared to temperate zones during growing seasons (Table SI.4)."*

**505: Is this 136350m3 the volume of the pools in this study? While it's small DOM**
350    **degradation in these sites, what would it be if scaling up for the whole Bouleau peatland? It may not only be 'slight effect' if considered collectively. In addition, seasonal variation in DOC concentration and degradability could also mean that in some months, the pools may act as 'hotspots' for GHG emission, which would be important information for peatland management along with global warming.**

355    As the objective was to evaluate the impact of DOM degradation in pools particularly, we did not scale it to the whole peatland surface. The DOM degradation in peatlands is driven by numerous other factors and mainly water table depth variations and is coupled with $CO_2$ and $CH_4$ dynamics. It is a larger process the research group will explore in a future paper.

However, the reviewer raised an interesting point. As degradation rates varying during the
360    growing season, it is an important element we considered it in the discussion (l. 533-537).

*"It is also important to note that the DOM in pools presents characteristics of recently produced DOM transferred from peat, its biodegradation might not affect importantly C from deeper peat horizons."*

**515: I may suggest an alternative next step to explore the effects from pools and peat**
365    **morphology on DOC transport from peat to water, as it is not so clearly assessed yet but could be important as regulating the water and DOC sources.**

This point was more detailed, and, in the conclusion, we propose a better coupling of both DOM and hydrological dynamics (l. 561-563).

370 *"As the dynamic of DOM in peat porewater seems closely connected to the hydrology of the peatland, it seems important to better connect it with the water sources and its circulation through the peat."*